# Latent-Guided Reasoning: Empowering Small LLMs with Large-Model Thinking

**Hanzhu Chen**[1][*][‡], **Lin Yang**[2][‡], **Jie Wang**[1][†], **Junhao Yan**[1], **Zhe Wang**[1],
**Xize Liang**[1], **Jianye Hao**[3]

[1]MoE Key Laboratory of Brain-inspired Intelligent Perception and Cognition,
University of Science and Technology of China
[2]Huawei Technologies Co., Ltd.
[3]College of Intelligence and Computing, Tianjin University
chenhz@mail.ustc.edu.cn,

## Abstract

Large Language Models (LLMs) have demonstrated remarkable capabilities in complex reasoning tasks, but their high computational costs limit their widespread practical application. We argue that this inefficiency arises from the tight coupling of high-level cognitive planning (devising the solution strategy) and low-level linguistic realization (generating step-by-step text). To address this challenge, we propose a novel collaborative framework that decouples these two processes through **Latent Guidance**. Our approach implements a division of labor: a large model acts as an **Implicit Thinker**, performing high-level cognitive planning and compressing its solution strategy into a set of compact latent guidance vectors. A small, efficient model then serves as an **Explicit Executor**, which receives this latent guidance to generate a concise and effective reasoning chain. This process is enabled by a dual-loss training objective, grounded in information-theoretic principles, where a reconstruction loss explicitly compels the latent guidance to become a high-fidelity representation of the full reasoning chain. Extensive experiments **on 8 diverse reasoning benchmarks** demonstrate that our method substantially enhances the reasoning capabilities of small models across various scales (from **0.5B to 8B**), allowing them to outperform strong baselines and exhibit superior generalization. Notably, our framework boosts small model accuracy by up to **13.9%** over its standalone baseline. Our work introduces a new, theoretically-grounded paradigm for empowering small models with large-model thinking, substantially improving the performance-cost trade-off for complex reasoning.

## 1 Introduction

The emergence of step-by-step reasoning techniques, such as Chain-of-Thought (CoT) (Wei et al., 2022; Kojima et al., 2022; Xia et al., 2025b) has enabled Large Language Models (LLMs) to demonstrate strong capabilities on complex multi-step reasoning problems. Due to its high degree of interpretability, step-by-step reasoning has been widely adopted in many applications that require detailed and verifiable solutions (Nye et al., 2021; Wang et al., 2022). However, the combination of massive parameter counts and long CoT generation processes leads to high computational costs (Achiam et al., 2023; Zhou et al., 2022; Xia et al., 2025a). This directly hinders their application in real-time or resource-constrained scenarios (Yin et al., 2023; Mei et al., 2025).

To address these high computational costs, existing research has primarily explored two directions. 1) **CoT Compression for Large Models** aims to shorten the reasoning chains or perform reasoning in a latent space to reduce generation costs (Goyal et al., 2023; Shen et al., 2025). However, these methods inevitably compromise the readability and interpretability of the original CoT, which is a significant drawback in fields where solution transparency is critical. 2) **Knowledge Distillation to Small Models** focuses on transferring reasoning capabilities from a large model to a smaller one,

---

[*]Work conducted during the research internship of Hanzhu Chen (chenhz@mail.ustc.edu.cn) at Huawei.
[‡]Equal contribution.
[†]Corresponding author. Email: jiewangx@ustc.edu.cn.

often via supervised fine-tuning (Chen et al., 2025) or more advanced optimization objectives (Hsieh et al., 2023; Li et al., 2024). While these methods can alleviate the cost issue, the small model's performance is often constrained by its limited parameter count, leading to poor generalization on complex, unseen reasoning tasks (Kang et al., 2023; Liao et al., 2025a). Therefore, **how to efficiently transfer the reasoning capabilities of large models to smaller counterparts, achieving both high performance and low inference cost, remains a critical challenge**.

To address this core challenge, we argue that this issue arises from a fundamental issue: the **tight coupling** of two distinct functions within the standard autoregressive framework. These are: 1) high-level **cognitive planning**, the process of devising a solution strategy, and 2) low-level **linguistic realization**, the generation of step-by-step text. In the standard reasoning paradigm, these two functions are intertwined, such that high-level cognitive planning can only be achieved through the medium of costly, step-by-step text generation.

In this work, we propose **Latent Guidance**, a new collaborative framework designed to empower small models with the advanced planning capabilities of large ones. At its core is a mechanism we term **Cognitive Distillation**, which decouples these two functions through a two-stage process. First, a large model, acting as an **Implicit Thinker**, is trained to generate a compact set of latent guidance vectors by processing special thought tokens. This training is guided by a dual-loss objective: a *task loss* ensures the cognitive plan correctly solves the problem, while a crucial *reconstruction loss* compels the latent vectors to encode the entire original reasoning chain. Subsequently, a small, efficient model, the **Explicit Executor**, receives this latent guidance and learns to generate a concise and effective textual solution. Our main contributions are as follows:

- **A Novel Framework for Efficient Reasoning.** We propose **Latent Guidance**, a framework empowering small models by decoupling cognitive planning from linguistic realization. Instead of distilling final text, our approach distills the high-level *solution strategy*, enabling a 7B model to gain up to a **13.9% accuracy** over its baseline, approaching the performance of much larger teacher models.

- **A Theoretically-Grounded Training Objective.** We introduce a dual-loss training objective where a reconstruction loss encourages the latent guidance is a high-fidelity representation of the full reasoning chain. Our formal analysis, grounded in information theory, provides explicit bounds connecting latent capacity to reasoning fidelity, supporting the formation of a robust and complete cognitive plan.

- **Comprehensive Empirical Validation.** Across **8** diverse reasoning benchmarks and multiple small models (**0.5B to 8B**), Latent Guidance consistently outperforms strong distillation baselines, demonstrating superior generalization. Furthermore, on a challenging long-form QA benchmark spanning over **12** domains, our method yields qualitatively superior explanations, highlighting its robust applicability to complex, unseen tasks.

- **In-depth Analysis of the Latent Cognitive Plan.** Through t-SNE visualizations and quantitative probing, we provide concrete evidence that latent guidance vectors organize into distinct clusters of high-level reasoning strategies. This analysis indicates the small model executes a structured, abstract plan, rather than just learning feature correlations.

## 2 RELATED WORK

### 2.1 KNOWLEDGE DISTILLATION FOR CHAIN-OF-THOUGHT REASONING

Distilling Chain-of-Thought (CoT) reasoning from large language models (LLMs) to smaller, more efficient ones is a significant research area. Approaches range from Supervised Fine-Tuning (SFT) on teacher-generated chains (Magister et al., 2023) to more advanced methods using distinct optimization objectives (Hsieh et al., 2023; Li et al., 2024), curriculum learning (Liang et al., 2025), cascaded learning stages (Dai et al., 2024), or blending neural networks with symbolic KB (Liao et al., 2025b; Bai et al., 2026). Furthermore, recent studies have explored enhancing reasoning capabilities through instantiated logical data (Wang et al., 2025). However, these methods can be viewed as a form of *outcome distillation*, training the small model to replicate the large model's final textual output. This often struggles by requiring the student to replicate a complex reasoning process for which it may lack the necessary parametric capacity or specialized knowledge (Kang et al., 2023; Liao et al., 2025a). In sharp contrast, our Latent Guidance framework introduces **Cognitive Distillation**. Instead of imitating the final text, our method distills the high-level *solution strategy* itself. The goal is not

for the small model to learn *how to reason* from scratch, but to learn the *linguistic realization* of a pre-computed cognitive plan. This shifts the burden of **cognitive planning** to the capable large model, allowing the small model to focus solely on the **linguistic realization** task.

## 2.2 REASONING IN LATENT SPACE

A distinct line of research explores latent-space reasoning to **accelerate a single large model's inference** by circumventing costly autoregressive text generation. Methods include iterative latent processing (Geiping et al., 2025; Kong et al., 2025), using placeholder tokens (e.g., <pause>) for added computation (Goyal et al., 2023; Pfau et al., 2024), or aligning explicit tokens with implicit states via self-distillation (Deng et al., 2023; Shen et al., 2025). Recently, foundational works like COCONUT (Hao et al., 2024) and LightThinker (Zhang et al., 2025) have validated the feasibility of performing reasoning in continuous latent space, often trading off some accuracy or interpretability for higher throughput and lower memory usage. Our work, however, repurposes latent computation for a different objective: **empowering a small model** by transferring the *cognitive planning* capabilities of a large one. While prior works focus on making large models more efficient, we focus on making small models more capable. This difference in purpose motivates our unique design. Crucially, unlike latent reasoning approaches that may lack strong supervision for the internal "thought" process, our framework's reconstruction loss ($\mathcal{L}_{\text{recon}}$) provides a powerful mechanism to ensure the latent guidance faithfully represents the full reasoning chain, yielding a robust, structured cognitive plan for the small model to execute.

## 3 METHODOLOGY

Our proposed **Latent Guidance** framework decouples the complex reasoning process into two specialized stages: **cognitive planning**, performed by a large model, and **linguistic realization**, handled by a small model. This section details the framework's architecture and the two-stage training procedure that enables this collaboration.

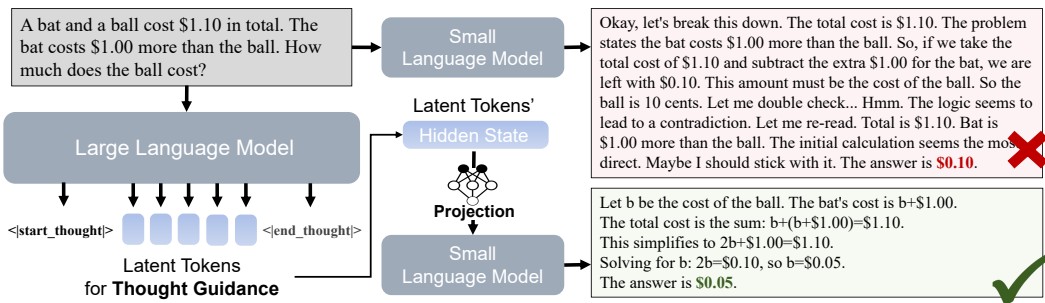

Figure 1: An overview of the Latent Guidance framework during inference. The large model (Implicit Thinker) generates compact latent guidance vectors from the input question by processing special *thought tokens*. The resulting hidden states are then passed through a projection layer to the small model (Explicit Executor). The small model, conditioned on both the question and the guidance, produces a concise and effective reasoning chain. As illustrated, without this guidance, the small model alone may generate a flawed reasoning path, highlighting the critical role of the latent guidance.

## 3.1 OVERALL FRAMEWORK

The core of our framework is the division of cognitive labor illustrated in Figure 1. The large model, acting as the **Implicit Thinker**, is trained to produce a sequence of compact latent vectors instead of generating lengthy text. These vectors, which we term *latent guidance*, serve as a high-level cognitive plan for the solution. The small model, acting as the **Explicit Executor**, then receives this guidance. Its sole task is to perform the linguistic realization of this plan, generating a step-by-step reasoning chain and the final answer. The entire process is enabled by a two-stage training strategy.

### 3.2 STAGE 1: TRAINING THE IMPLICIT THINKER TO FORMULATE A COGNITIVE PLAN

The primary objective of this stage is to train the large model (Implicit Thinker) to take a question $Q$ as input and produce a sequence of information-rich latent vectors that encode its solution strategy. To create explicit anchor points for these latent representations, we introduce a set of special thought tokens: '<start_thought>', '<end_thought>', and $K$ placeholder tokens '$\{< thought_1 >, \ldots, < thought_K >\}$'.

Given a training example consisting of a question $Q$, a reasoning chain $R = (r_1, \ldots, r_M)$, and an answer $A = (a_1, \ldots, a_L)$, we reformat the target sequence. Instead of predicting the full CoT, the large model is trained on a template that replaces the reasoning text with our latent tokens: '$< start\_thought >< thought_1 > \cdots < thought_K >< end\_thought > A$'.

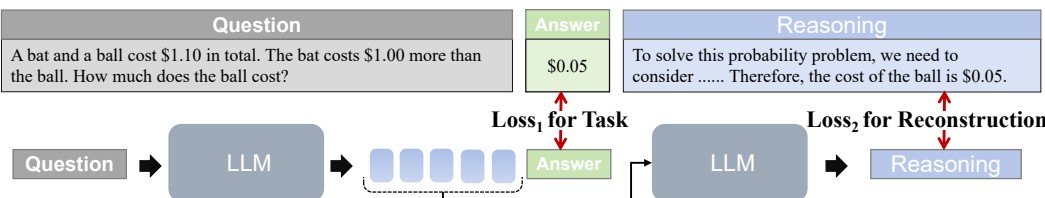

Figure 2: The dual-loss training for the Implicit Thinker (large model). The model is optimized on two objectives simultaneously: a task loss to ensure the correctness of the cognitive plan and a reconstruction loss to compel the latent vectors to encode the full reasoning chain.

To ensure the hidden states corresponding to these tokens meaningfully encode the complete solution strategy, we employ a **dual-loss training objective**, as shown in Figure 2:

$$\mathcal{L}_{\text{LLM}} = \mathcal{L}_{\text{task}} + \mathcal{L}_{\text{recon}}. \tag{1}$$

.

**Task Loss ($\mathcal{L}_{\text{task}}$): Grounding the Plan in Correctness.** This is a standard autoregressive language modeling loss focused solely on predicting the final answer $A$. This loss ensures that the model's cognitive plan is grounded in the ultimate goal of solving the problem correctly. Let $\theta_{\text{LLM}}$ be the parameters of the large model. The task loss is:

$$\mathcal{L}_{\text{task}} = -\sum_{j=1}^{L} \log P(a_j \mid Q, \{\text{thought}_k\}_{k=1}^{K}, a_{<j}; \theta_{\text{LLM}}). \tag{2}$$

**Reconstruction Loss ($\mathcal{L}_{\text{recon}}$): Encoding the Reasoning Chain.** This loss is the cornerstone of our method. After a forward pass, we extract the last-layer hidden states $\{\mathbf{h}_1, \ldots, \mathbf{h}_K\}$ corresponding to the thought tokens. These states form our latent guidance, $\mathbf{H}_{\text{guidance}}$. The reconstruction loss then compels these latent vectors to contain sufficient information to fully reconstruct the original reasoning chain $R$:

$$\mathcal{L}_{\text{recon}} = -\sum_{i=1}^{M} \log P(r_i \mid \mathbf{H}_{\text{guidance}}, r_{<i}; \theta_{\text{LLM}}). \tag{3}$$

**This objective is not just empirically effective; we provide a principled information-theoretic analysis in Appendix A**. Specifically, minimizing $\mathcal{L}_{\text{recon}}$ is equivalent to maximizing the mutual information between $\mathbf{H}_{\text{guidance}}$ and the reasoning chain $R$, i.e.,

$$I(R; \mathbf{H}_{\text{guidance}} \mid Q) \geq H(R \mid Q) - \mathcal{L}_{\text{recon}}. \tag{4}$$

This compels that the guidance encodes a complete and faithful representation of the cognitive plan.

### 3.3 STAGE 2: TRAINING THE EXPLICIT EXECUTOR FOR LINGUISTIC REALIZATION

Once the Implicit Thinker is trained, we use it to generate and store the latent guidance vectors $\mathbf{H}_{\text{guidance}}$ for every problem in the training set. The goal of this second stage is to train the small model (Explicit Executor) to take the question $Q$ and the latent guidance as input, and generate the full, human-readable reasoning chain and answer $(R, A)$.

**Bridging Latent Spaces with a Projection Layer.** Since the large and small models may have different hidden dimensions, their latent spaces are incompatible. We therefore introduce a lightweight projection layer to bridge the two spaces. In practice, this is implemented as:

$$\mathbf{H}'_{\text{guidance}} = \text{MLP}(\mathbf{H}_{\text{guidance}}), \tag{5}$$

where the MLP consists of two linear layers with an intermediate dimension of 2048, a GELU activation, a dropout layer with rate 0.1, and a final LayerNorm. This design provides both capacity and regularization, ensuring stable transfer of information.

**Training Objective: Realizing the Cognitive Plan.** The small model is then fine-tuned with a standard language modeling objective. It learns to generate the complete solution sequence $(R, A)$ conditioned on both the question and the projected guidance vectors. Let $\theta_{\text{SLM}}$ be the parameters of the small model, the loss is:

$$\mathcal{L}_{\text{SLM}} = -\sum_{i=1}^{M+L} \log P(t_i \mid Q, \mathbf{H}'_{\text{guidance}}, t_{<i}; \theta_{\text{SLM}}), \tag{6}$$

where $(t_1, \ldots, t_{M+L})$ is the concatenated token sequence of the reasoning chain $R$ and the answer $A$. This training effectively teaches the small model to perform linguistic realization on the cognitive plan provided by the large model.

### 3.4 Decoupled Inference Process

During inference, the framework operates in a highly efficient, two-step manner:

1. **Cognitive Planning**: Given a new question, the trained large model (Implicit Thinker) performs a single forward pass to generate the latent guidance vectors $\mathbf{H}_{\text{guidance}}$. This step is extremely fast as it avoids any autoregressive text generation.

2. **Linguistic Realization**: The question and the generated $\mathbf{H}_{\text{guidance}}$ are passed to the trained small model (Explicit Executor), which then autoregressively generates the final reasoning chain and answer.

This decoupled process leverages the large model's powerful planning abilities while capitalizing on the small model's efficiency for text generation, achieving a superior balance of reasoning accuracy and computational efficiency.

## 4 Experiments

To comprehensively validate our proposed method, we structure our experimental evaluation to systematically demonstrate our framework's effectiveness, generalization, and core mechanisms. Our experiments are organized as follows:

- First, we establish the **Broad Effectiveness** of our framework, showing that it consistently improves reasoning performance across a wide range of models (from 0.5B to 8B) on **8 diverse reasoning benchmarks** against strong contemporary baselines (Sec 4.2).

- Second, we demonstrate the **Superior Generalization** of our method. We move beyond standard accuracy metrics to show that our method produces qualitatively superior and more concise explanations on a challenging long-form QA benchmark that spans **over 12 distinct domains** (Sec 4.3).

- Third, we conduct **In-depth Mechanism Analysis** to dissect the principles underlying these results. Through ablation studies, visualizations, and quantitative probing, we provide direct evidence that the framework learns a structured, abstract cognitive plan rather than superficial feature correlations (Sec 4.4).

To supplement these findings, we provide extensive appendices that substantiate our core claims. This begins with a **formal theoretical analysis** (Appendix A), grounded in information theory, rate-distortion theory, and Fano's inequality, which justifies our training objective and proves the robustness of the learned cognitive plan. To empirically validate these theories, we then provide **extensive diagnostics** (Appendix B), including neural mutual information estimates, latent capacity scaling experiments, and decoder robustness measurements. Notably, these diagnostics confirm that our latent guidance captures a substantial amount of quantifiable information about the complete reasoning chain, estimated at approximately **3.1 nats**.

## 4.1 EXPERIMENTAL SETUP

**Datasets** To rigorously evaluate performance and generalization, we partition our benchmarks into In-Domain (ID) and Out-of-Domain (OOD) categories. For ID training, we use the splits from **GSM8K** (Cobbe et al., 2021) and **BBH** (Suzgun et al., 2022), evaluating on their respective test sets. To assess OOD generalization, we test on a wide array of unseen datasets without task-specific fine-tuning: **AGIEval** (Zhong et al., 2023), **ARC** (Clark et al., 2018) (easy and challenge), **Odyssey-Math** (Netmind.AI, 2024), **SVAMP** (Patel et al., 2021), and **AQuA** (Ling et al., 2017). We use **ELI5-Test** (Fan et al., 2019) for qualitative evaluation of generated rationales.

**Models and Baselines** Our experiments utilize open-source models for reproducibility, with **Qwen2.5-32B-Instruct** (Hui et al., 2024) as the large model (Implicit Thinker). For small models (Explicit Executors), we evaluate on a range of base models (**LLaMA2-7B** (Touvron et al., 2023), **LLaMA-3-8B** (Dubey et al., 2024), **Qwen2-0.5B/1.5B/7B** (Team, 2024)) and use the powerful **Qwen2.5-7B-Instruct** (Hui et al., 2024) for deeper analysis. We compare against a comprehensive suite of baselines including SFT (Supervised Fine-Tuning), Knowledge Distillation (KD), and strong contemporary methods: **Std-CoT** (Magister et al., 2023), **MT-CoT** (Li et al., 2024), **Step-by-step** (Hsieh et al., 2023), **KARD** (Kang et al., 2023), **CasCoD** (Dai et al., 2024), and **NesyCD** (Liao et al., 2025b). Our focus on distillation-based baselines ensures a fair comparison. Please refer to Appendix D for detailed experimental setup.

## 4.2 BROAD EFFECTIVENESS ACROSS DIVERSE MODELS

To establish the general applicability of our approach, we first evaluate its effectiveness across various model families and scales against established distillation methods. The results, presented in Table 1, show that Latent Guidance consistently improves the reasoning performance of small models. For example, when applied to LLaMA-3-8B, our framework achieves an overall average accuracy of **73.1%**, an increase of 2.8 percentage points over the strongest competing baseline, NesyCD (70.3%). This performance gain is particularly driven by its superior generalization to OOD datasets, where our method achieves an OOD average of **71.9%**, significantly outperforming NesyCD's 68.1%. This trend is consistent across other model families; with Qwen2-7B, our method leads the OOD average by 3.6 points. These findings support our central hypothesis that decoupling cognitive planning from linguistic realization is a broadly effective strategy for enhancing small models' reasoning capabilities. Full results for LLaMA2-7B are available in Appendix C.

## 4.3 ANALYSIS OF REASONING QUALITY AND GENERALIZATION

Having established the framework's quantitative improvements, we now investigate the drivers behind its strong out-of-domain (OOD) performance. We analyze both the quantitative efficiency and qualitative richness of the generated reasoning chains, using the powerful Qwen2.5-7B-Instruct as the small model.

**Quantitative Performance and Generalizable Conciseness** Table 2 reveals two key findings that explain our method's OOD success. First, Latent Guidance consistently boosts accuracy on challenging OOD benchmarks like Odyssey-Math (+7.2% over KD). Second, it produces substantially more concise reasoning chains, reducing the token count compared to SFT while simultaneously improving accuracy. This demonstrates that the high-level cognitive plan enables the small model to generate more focused and *generalizable* reasoning paths, avoiding the verbose, exploratory steps often seen in less-guided generation on unseen problems. The model learns to execute a direct strategy rather than overfitting to the stylistic artifacts of the training data.

**Qualitative Proof of Broad Generalization** We further test our framework's generalization on the completely OOD, long-form question-answering benchmark, ELI5-Test. As detailed in Table 3, the results show a clear and consistent advantage. Across **more than 12 diverse domains**, from Biology to Economics, our method's outputs are judged by GPT-4o as substantially superior in both **Correctness** and **Relevance**, consistently outperforming both SFT and KD. This strong performance across numerous unseen topics demonstrates that the distilled cognitive plan is not task-specific but imparts a robust, broadly applicable reasoning structure. The small model is not just producing more accurate answers; it is generating qualitatively superior and more insightful explanations, confirming the powerful generalization capability of our framework.

| Methods | In-Domain | | Out-Of-Domain | | | | Overall Avg. |
|---|---|---|---|---|---|---|---|
| | BBH-test | GSM8K | AGIEval | ARC-E | ARC-C | OOD Avg. | |
| *# LLaMA-3-8B based* | | | | | | | |
| Std-CoT | 79.4 | 61.6 | 41.3 | 83.2 | 71.9 | 65.5 | 67.5 |
| MT-CoT | 62.8 | 13.1 | 43.9 | 83.6 | 72.3 | 66.6 | 55.1 |
| Step-by-step | 64.0 | 11.5 | 43.7 | 84.3 | 74.6 | 67.5 | 55.6 |
| KARD (BM25) | 81.4 | 64.3 | 43.4 | 85.6 | 76.1 | 68.4 | 70.2 |
| CasCoD | 32.1 | 59.1 | 23.6 | 34.6 | 27.7 | 28.6 | 35.4 |
| NesyCD | 82.2 | 64.9 | 44.1 | 84.7 | 75.4 | 68.1 | 70.3 |
| Ours | **82.5** | **67.4** | **45.2** | **91.8** | **78.6** | **71.9** | **73.1** |
| *# Qwen2-0.5B based* | | | | | | | |
| Std-CoT | 65.8 | 26.7 | 25.6 | 43.6 | 32.0 | 33.7 | 38.7 |
| MT-CoT | 47.2 | 5.3 | 27.7 | 46.0 | 35.1 | 36.3 | 32.3 |
| Step-by-step | 44.2 | 5.2 | 26.2 | 41.8 | 36.2 | 34.7 | 30.7 |
| KARD (BM25) | 66.3 | 30.9 | 23.9 | 48.9 | 37.2 | 36.7 | 41.4 |
| CasCoD | 37.6 | 27.7 | 15.6 | 21.5 | 14.8 | 17.3 | 23.4 |
| NesyCD | 68.7 | 32.2 | 28.4 | 46.8 | 36.7 | 37.3 | 42.6 |
| Ours | **71.0** | **34.8** | **29.1** | **49.1** | **37.6** | **38.6** | **44.3** |
| *# Qwen2-1.5B based* | | | | | | | |
| Std-CoT | 68.2 | 52.7 | 34.0 | 69.3 | 56.4 | 53.2 | 56.1 |
| MT-CoT | 58.0 | 6.7 | 34.2 | 72.7 | 57.5 | 54.8 | 45.8 |
| Step-by-step | 48.4 | 5.8 | 34.4 | 72.1 | 57.6 | 54.7 | 43.7 |
| KARD (BM25) | 72.2 | 55.4 | 31.2 | 74.0 | 62.2 | 55.8 | 59.0 |
| CasCoD | 31.7 | 53.4 | 24.7 | 57.1 | 47.8 | 43.2 | 42.9 |
| NesyCD | 74.6 | 55.8 | 35.1 | 73.6 | 58.2 | 55.6 | 59.5 |
| Ours | **78.4** | **56.0** | **36.1** | **77.2** | **60.6** | **58.0** | **61.7** |
| *# Qwen2-7B based* | | | | | | | |
| Std-CoT | 80.7 | 71.5 | 49.9 | 90.5 | 80.3 | 73.6 | 74.6 |
| MT-CoT | 70.0 | 15.2 | 49.4 | 90.9 | 80.2 | 73.5 | 61.1 |
| Step-by-step | 68.8 | 15.2 | 49.1 | 72.1 | 71.8 | 64.3 | 55.4 |
| KARD (BM25) | 80.2 | 75.3 | 49.6 | 92.1 | 83.5 | 75.1 | 76.1 |
| CasCoD | 35.7 | 72.3 | 37.4 | 70.1 | 63.1 | 56.9 | 55.7 |
| NesyCD | 80.9 | 76.3 | 49.9 | 91.9 | 82.9 | 74.9 | 76.4 |
| Ours | **82.1** | **77.5** | **54.4** | **94.8** | **86.3** | **78.5** | **79.0** |

Table 1: Performance (%) of different methods on various small base models across five reasoning benchmarks. The **OOD Avg.** column highlights generalization performance. **Bold** indicates the best performance among distillation-based methods for each model. Our Latent Guidance framework consistently outperforms baselines.

Table 2: Comprehensive Performance Evaluation using an instruct-tuned Qwen2.5-7B small model. The table shows quantitative accuracy (%) and average token counts on reasoning benchmarks.

| Type | Dataset / Category | Accuracy (%) | | | Avg. Tokens | | |
|---|---|---|---|---|---|---|---|
| | | SFT | KD | Ours | SFT | KD | Ours |
| *Quantitative Benchmark Performance* | | | | | | | |
| In-Domain | GSM8K | 77.3 | 73.0 | **80.5** | 235.4 | 253.7 | **128.9** |
| In-Domain | BBH | 79.2 | 77.6 | **79.8** | 335.3 | 337.8 | **192.9** |
| Out-of-Domain | AGIEval | 47.1 | 55.8 | **56.7** | 430.8 | 537.6 | **211.2** |
| Out-of-Domain | ARC-E | 94.5 | 89.1 | **96.4** | 230.3 | 261.5 | **122.3** |
| Out-of-Domain | ARC-C | 89.4 | 83.2 | **90.9** | 252.3 | 294.4 | **138.3** |
| Out-of-Domain | Odyssey-Math | 15.5 | 17.9 | **22.7** | 522.2 | 563.1 | **343.7** |
| Out-of-Domain | SVAMP | 83.0 | 82.9 | **85.9** | 155.1 | 171.8 | **68.6** |
| Out-of-Domain | AQuA | 65.4 | 69.5 | **71.7** | 316.2 | 427.3 | **186.4** |

Table 3: Detailed GPT-4o evaluation results on the ELI5-Test dataset, focusing on core reasoning quality. We report scores (1-10) for Correctness (Corr.) and Relevance (Rel.). Our method consistently produces more correct and relevant explanations.

| Category | SFT | | | KD | | | Ours | | |
|---|---|---|---|---|---|---|---|---|---|
| | Corr. | Rel. | Avg | Corr. | Rel. | Avg | Corr. | Rel. | Avg |
| Biology | 7.65 | 8.20 | 7.93 | 7.90 | 8.10 | 8.00 | 8.25 | 8.65 | 8.45 |
| Chemistry | 7.60 | 8.20 | 7.90 | 7.95 | 8.05 | 8.00 | 8.45 | 8.65 | 8.55 |
| Culture | 7.95 | 8.20 | 8.08 | 8.40 | 8.60 | 8.50 | 8.50 | 8.60 | 8.55 |
| Earth Science | 7.45 | 7.70 | 7.58 | 7.85 | 8.25 | 8.05 | 8.75 | 8.75 | 8.75 |
| Economics | 8.05 | 8.25 | 8.15 | 8.10 | 8.30 | 8.20 | 8.75 | 9.00 | 8.88 |
| Engineering | 7.90 | 8.35 | 8.13 | 8.35 | 8.70 | 8.53 | 8.40 | 8.85 | 8.63 |
| Mathematics | 8.45 | 8.50 | 8.48 | 8.20 | 8.40 | 8.30 | 8.95 | 9.15 | 9.05 |
| Other | 7.65 | 7.80 | 7.73 | 7.85 | 8.05 | 7.95 | 8.40 | 8.60 | 8.50 |
| Physics | 7.60 | 8.20 | 7.90 | 8.10 | 8.35 | 8.23 | 8.55 | 8.95 | 8.75 |
| Psychology | 8.45 | 8.65 | 8.55 | 7.80 | 8.05 | 7.93 | 8.75 | 9.20 | 8.98 |
| Repost | 7.60 | 7.90 | 7.75 | 8.05 | 8.35 | 8.20 | 8.65 | 8.65 | 8.65 |
| Technology | 8.25 | 8.55 | 8.40 | 7.95 | 8.40 | 8.18 | 8.80 | 9.15 | 8.98 |

## 4.4 Unpacking the Mechanism of Latent Guidance

To better understand *why* our framework is effective, we now analyze its internal mechanics. We conduct an ablation study to validate our core design choices and use visualizations to gain insight into the learned cognitive plan.

**Ablation Study: Validating Core Design Choices**   We performed an ablation study to isolate the impact of our core design choices: the reconstruction loss ($\mathcal{L}_{\text{recon}}$), the number of thought tokens ($K$), and the architecture of the projection layer. As shown in Table 4, removing the reconstruction loss leads to a clear performance degradation across all datasets (e.g., -2.3% on GSM8K), confirming its critical role in ensuring the latent guidance faithfully encodes the cognitive plan. The study on the number of thought tokens shows that reducing to $K = 3$ results in a slight decline, while increasing to $K = 10$ yields only marginal gains, suggesting our default of $K = 5$ is a reasonable trade-off.

The ablation on the projection layer architecture further validates our design. Replacing our 2-layer MLP with a simpler 1-layer linear projection leads to a consistent performance drop (e.g., -1.5% on GSM8K). This suggests that a simple linear mapping is insufficient to fully bridge the two models' latent spaces, creating an information bottleneck. Conversely, a more complex 3-layer MLP provides no meaningful improvement and even results in a slight performance decrease on some datasets, likely due to the additional capacity leading to overfitting on the training data. This strongly indicates diminishing returns and reinforces our choice of the 2-layer design. These results demonstrate that our proposed configuration effectively balances model capacity and performance.

Table 4: Ablation study on the key components of our framework using the Qwen2.5-7B-Instruct small model. Performance is reported as accuracy (%). Our default 2-layer MLP projection with K=5 proves to be a well-balanced configuration.

| Configuration | GSM8K | SVAMP | ARC-C |
|---|---|---|---|
| **Full model (2-layer MLP, K=5)** | **80.5** | **85.9** | **90.9** |
| *Ablation on Core Loss and Tokens* | | | |
| w/o Reconstruction Loss ($\mathcal{L}_{\text{recon}}$) | 78.2 | 83.9 | 89.5 |
| Fewer thought tokens (K=3) | 79.4 | 85.5 | 89.8 |
| More thought tokens (K=10) | 81.1 | 86.5 | 91.4 |
| *Ablation on Projection Layer* | | | |
| Simple Projection (1-layer MLP) | 79.0 | 84.1 | 89.5 |
| Deeper Projection (3-layer MLP) | 80.4 | 84.4 | 89.8 |

**Visualizing the Structure of the Cognitive Plan**   The ablation study confirms the functional importance of our training objective. To visually inspect what this objective imparts to the latent space, we visualized the $\mathbf{H}_{\text{guidance}}$ vectors from the GSM8K test set using t-SNE (Figure 3). The visualization reveals that the vectors form distinct clusters without any explicit clustering supervision.

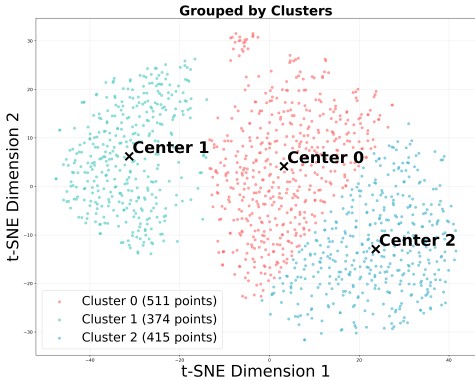

Figure 3: t-SNE visualization of the latent guidance vectors for the GSM8K test set.

| Probing Task | Acc$_L$ | Acc$_R$ |
|---|---|---|
| *Reasoning Step Count Prediction* | | |
| Exact Step Match | 36.0 | 26.3 |
| Tolerance $\pm 1$ Step | 74.4 | 50.5 |
| Tolerance $\pm 2$ Steps | 94.2 | 76.9 |
| *Mathematical Operator Prediction* | | |
| Presence of '+' (Binary) | 85.4 | 50.0 |
| Presence of '-' (Binary) | 87.0 | 50.0 |
| Presence of '*' (Binary) | 84.5 | 50.0 |
| Presence of '/' (Binary) | 82.1 | 50.0 |
| Full Operator Set (Multi-class) | 54.7 | 14.6 |

Table 5: Accuracy (%) of a probing MLP. 'Acc$_L$' and 'Acc$_R$' denote accuracy from our guidance (Latent) and a random baseline, respectively.

A qualitative review of samples within these clusters showed a strong semantic coherence, with each cluster corresponding to a different high-level reasoning paradigm:

- **Cluster 1 (Sequential Multi-Step Reasoning):** Problems requiring a sequence of dependent calculations (e.g., tracking a value over time).
- **Cluster 2 (Direct Calculation & Aggregation):** Problems solvable by performing parallel calculations on independent inputs followed by a final aggregation.
- **Cluster 3 (Relational & Constraint-Based Modeling):** Problems defined by algebraic or logical relationships that require setting up and solving equations.

This emergent structure provides strong evidence for our hypothesis that the latent guidance represents a high-level, structured cognitive plan, rather than an unstructured feature representation. This offers a compelling explanation for our framework's success: the small model is not merely mimicking token patterns but is executing a pre-computed, well-structured cognitive plan.

**Probing for Semantic Structure in Latent Guidance**  While the t-SNE visualization suggests a high-level organization of reasoning strategies, we conducted a series of probing tasks to quantitatively verify whether the latent guidance vectors, $\mathbf{H}_{\text{guidance}}$, encode specific, fine-grained semantic information. We trained a simple Multi-Layer Perceptron (MLP) classifier directly on the frozen latent guidance vectors to predict distinct properties of the ground-truth reasoning chain. We compare its performance against a random baseline, where an identical MLP is trained on randomly initialized vectors of the same dimension. The results, presented in Table 5, offer compelling evidence that the latent guidance is semantically rich. Predicting complex properties like the exact number of reasoning steps is inherently challenging, yet our probing model achieves a notable 0.360 accuracy, significantly outperforming the 0.263 baseline. The model's grasp of overall reasoning complexity is even more evident when allowing for minor tolerance, reaching 0.744 accuracy for $\pm 1$ step and an impressive 0.942 for $\pm 2$ steps—both substantially higher than their respective baselines. This quantitative analysis strongly suggests that the large model, guided by the reconstruction loss, learns to embed a structured and detailed cognitive plan into the latent vectors, capturing not just the general strategy but also specific procedural elements required for the solution.

## 5 CONCLUSION

We introduced Latent Guidance, a novel framework that empowers small language models with the advanced reasoning of large ones through **Cognitive Distillation**. Grounded in information theory, our dual-loss objective ensures the distilled guidance is a high-fidelity representation of the teacher's reasoning process. By decoupling high-level cognitive planning from linguistic realization, our method significantly improves the performance-cost trade-off for complex reasoning. Extensive experiments demonstrate that our framework not only yields substantial accuracy gains with strong OOD generalization but also produces more concise and qualitatively superior reasoning chains. Our in-depth analysis, using t-SNE visualizations and probing tasks, provides direct evidence that our method successfully distills a structured, high-level cognitive plan, rather than superficial token patterns. This work opens a promising new direction for achieving efficient, high-fidelity reasoning through specialized collaboration between large and small models.

ACKNOWLEDGMENTS

This work was supported in part by National Key R&D Program of China under contract 2022ZD0119801, National Nature Science Foundations of China grants U23A20388 and 62021001. We would like to thank all the anonymous reviewers for their insightful comments.

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

# A  THEORETICAL ANALYSIS AND PROOFS

This appendix supplies a self-contained theoretical perspective and detailed proofs that underpin the *Latent Guidance* framework presented in the main text. Our analysis builds directly on classical results in information theory and statistical learning, including cross-entropy decomposition and KL non-negativity (Cover & Thomas, 2006), Fano's inequality (Cover & Thomas, 2006; Yu, 1997), rate–distortion theory (Berger, 1971), the data-processing inequality (Cover & Thomas, 2006), and robustness bounds based on Lipschitz continuity and Pinsker's inequality (Xu & Mannor, 2012; Villani et al., 2008). We also draw on covering-number arguments for continuous reasoning spaces (Haussler & Opper, 1997), as well as recent neural mutual-information estimators such as MINE (Belghazi et al., 2018) and InfoNCE (Oord et al., 2018).

Our goals are fourfold:

1. Connect the reconstruction objective to mutual information between the chain-of-thought and the latent guidance (a precise information-theoretic lower bound).

2. Provide an exact-recovery (Fano-style) lower bound illustrating how a small reconstruction error forces the latent guidance to carry nearly the full reasoning information.

3. Prove robustness bounds showing that small perturbations (noise or quantization) to the latent guidance induce only bounded changes in the decoder distribution; and discuss the projection (data-processing) bottleneck.

4. Close the loop by giving *explicit, empirically grounded* upper bounds and practical clarifications on constants (e.g., Lipschitz constants, covering numbers) so that the full range of latent-capacity requirements is sandwich-bounded and experimentally verifiable.

## A.1  NOTATION AND SETUP

Let $(Q, R)$ denote a data pair where $Q$ is a problem statement (context) and $R = (r_1, \ldots, r_T)$ is the corresponding chain-of-thought (CoT), modeled as a token sequence of length $T$ (for brevity we use $T$ to denote the typical reasoning-chain length). The Implicit Thinker (teacher) produces a latent guidance $H \in \mathbb{R}^{K \times d}$ (a sequence of $K$ vectors in $\mathbb{R}^d$). The decoder/Explicit Executor's reconstruction model is denoted $q_\theta(R \mid H, Q)$; the training reconstruction loss is

$$\mathcal{L}_{\text{recon}}(\theta) \ = \ \mathbb{E}_{P(Q,R,H)}\big[-\log q_\theta(R \mid H, Q)\big],$$

where $P(Q, R, H)$ is the joint distribution induced by the data and the teacher encoder (we allow the encoder to be deterministic or stochastic). We use standard information-theoretic quantities:

$$H(R \mid Q), \quad H(R \mid H, Q), \quad I(R; H \mid Q) = H(R \mid Q) - H(R \mid H, Q).$$

## A.2  MUTUAL INFORMATION LOWER BOUND FROM RECONSTRUCTION LOSS

The next theorem makes explicit the connection between the reconstruction loss and the conditional mutual information between the reasoning chain and the latent guidance.

**Theorem A.1** (Mutual-information lower bound via reconstruction loss)**.** *For any decoder distribution* $q_\theta(R \mid H, Q)$ *it holds that*

$$I(R; H \mid Q) \ \geq \ H(R \mid Q) - \mathcal{L}_{\text{recon}}(\theta). \tag{7}$$

*Equivalently, if* $\mathcal{L}_{\text{recon}}(\theta) \leq H(R \mid Q) - \delta$, *then* $I(R; H \mid Q) \geq \delta$.

*Proof.* This follows directly from the variational representation of cross-entropy and the non-negativity of KL divergence (see e.g., Cover & Thomas, 2006).

By definition of conditional mutual information,

$$I(R; H \mid Q) = H(R \mid Q) - H(R \mid H, Q).$$

We relate $\mathcal{L}_{\text{recon}}(\theta)$ to $H(R \mid H, Q)$. For fixed $(H, Q)$, let $P_{R \mid H, Q}$ denote the true conditional distribution of $R$ given $(H, Q)$. Then for any candidate decoder $q_\theta(\cdot \mid H, Q)$ the decomposition using KL divergence gives

$$\mathbb{E}_{P(R \mid H,Q)}[-\log q_\theta(R \mid H, Q)] \ = \ H\big(P_{R \mid H,Q}\big) + \text{KL}\big(P_{R \mid H,Q} \,\|\, q_\theta(\cdot \mid H, Q)\big).$$

Taking expectation over $(H, Q) \sim P(H, Q)$ yields

$$\mathcal{L}_{\text{recon}}(\theta) \;=\; \mathbb{E}_{P(H,Q)}\Big[H\big(P_{R|H,Q}\big)\Big] + \mathbb{E}_{P(H,Q)}\Big[\text{KL}\big(P_{R|H,Q} \,\|\, q_\theta(\cdot \mid H, Q))\big)\Big].$$

Since KL is non-negative, we have

$$\mathcal{L}_{\text{recon}}(\theta) \;\geq\; \mathbb{E}_{P(H,Q)}\Big[H\big(P_{R|H,Q}\big)\Big] \;=\; H(R \mid H, Q).$$

Combine with the mutual information identity to obtain

$$I(R; H \mid Q) = H(R \mid Q) - H(R \mid H, Q) \geq H(R \mid Q) - \mathcal{L}_{\text{recon}}(\theta),$$

which is equation 7. $\qquad\square$

*Remark* A.2. This theorem shows that minimizing $\mathcal{L}_{\text{recon}}$ (with a sufficiently expressive decoder) forces $H$ to carry more mutual information about $R$. In words: a low reconstruction loss is a certificate that $H$ encodes (nearly) the full chain-of-thought information.

### A.3  FANO-STYLE EXACT-RECOVERY LOWER BOUND

The following Fano-type bound shows that if the decoder can reconstruct the chain-of-thought with high probability, then $I(R; H \mid Q)$ must be large; this gives a bit-level intuition / necessary condition relating latent capacity to the CoT's information content.

**Theorem A.3** (Fano-type lower bound). *Assume $R$ takes values in a finite set $\mathcal{R}$. Suppose there exists a deterministic decoding rule $\widehat{R}(H, Q)$ such that the exact reconstruction probability satisfies $\Pr[\widehat{R}(H, Q) \neq R] \leq \varepsilon$. Then*

$$I(R; H \mid Q) \;\geq\; H(R \mid Q) - h(\varepsilon) - \varepsilon \log(|\mathcal{R}| - 1), \tag{8}$$

*where $h(\varepsilon) = -\varepsilon \log \varepsilon - (1 - \varepsilon) \log(1 - \varepsilon)$ is the binary entropy function (natural logarithms) and $|\mathcal{R}|$ denotes the cardinality of the set of possible reasoning chains.*

*Proof.* This is a direct application of Fano's inequality (see Cover & Thomas, 2006; Yu, 1997).
Define the error indicator random variable

$$E \;=\; \mathbf{1}\{\widehat{R}(H, Q) \neq R\}.$$

Note $E$ is a (deterministic) function of $(R, H, Q)$, hence $H(E \mid R, H, Q) = 0$. Consider the conditional entropy $H(R \mid H, Q)$. Using the chain rule:

$$H(R \mid H, Q) = H(E, R \mid H, Q) - H(E \mid R, H, Q) = H(E \mid H, Q) + H(R \mid E, H, Q),$$

where we used $H(E \mid R, H, Q) = 0$. Now expand the second term by conditioning on $E$:

$$H(R \mid E, H, Q) = \Pr(E = 0 \mid H, Q)\, H(R \mid E = 0, H, Q) + \Pr(E = 1 \mid H, Q)\, H(R \mid E = 1, H, Q).$$

When $E = 0$ we have exact reconstruction, i.e. $R = \widehat{R}(H, Q)$ and so $H(R \mid E = 0, H, Q) = 0$. Therefore

$$H(R \mid E, H, Q) = \Pr(E = 1 \mid H, Q)\, H(R \mid E = 1, H, Q).$$

But conditional entropy is bounded by the log-cardinality, $H(R \mid E = 1, H, Q) \leq \log(|\mathcal{R}| - 1)$ (since when $E = 1$ the possible values for $R$ exclude at least the decoded value and are at most $|\mathcal{R}| - 1$). Hence

$$H(R \mid H, Q) \leq H(E \mid H, Q) + \Pr(E = 1 \mid H, Q) \log(|\mathcal{R}| - 1).$$

Taking expectation over $(H, Q)$ gives

$$H(R \mid H, Q) \leq H(E) + \Pr(E = 1) \log(|\mathcal{R}| - 1).$$

By assumption $\Pr(E = 1) \leq \varepsilon$ and the entropy of a Bernoulli variable with mean $\Pr(E = 1)$ is at most $h(\varepsilon)$. Thus

$$H(R \mid H, Q) \leq h(\varepsilon) + \varepsilon \log(|\mathcal{R}| - 1).$$

Combining with $I(R; H \mid Q) = H(R \mid Q) - H(R \mid H, Q)$ yields the desired bound equation 8. $\quad\square$

*Remark* A.4. The Fano bound gives a necessary condition in terms of bits: to achieve small error $\varepsilon$, the latent guidance must capture nearly $H(R \mid Q)$ bits (up to the $h(\varepsilon) + \varepsilon \log(|\mathcal{R}| - 1)$ slack). For continuous or very large $\mathcal{R}$, one uses other distortion-oriented analyses (see rate–distortion below).

### A.4 RATE–DISTORTION PERSPECTIVE (INFORMAL BUT CONSTRUCTIVE)

This viewpoint follows the classical rate–distortion framework (Berger, 1971; Cover & Thomas, 2006). The rate–distortion viewpoint complements the previous bounds by asking: *what is the minimum mutual information $I(R; H \mid Q)$ required to ensure an expected distortion (reconstruction loss) not exceeding $D$?* Formally, for a distortion measure $d(R, \widehat{R})$ (we may take $d = -\log q_\theta(R \mid H, Q)$ as a probabilistic distortion), the conditional rate–distortion function is defined as

$$R_{R|Q}(D) = \inf_{P(H|R,Q):\, \mathbb{E}[d(R,\widehat{R})] \leq D} I(R; H \mid Q).$$

In particular, in the limit of vanishing distortion the minimum rate approaches $H(R \mid Q)$. Although deriving closed-form $R_{R|Q}(D)$ is in general intractable for natural-language CoTs, the conceptual implication is that the latent capacity (measured in bits, approximated by $K \cdot d$) must exceed the rate $R_{R|Q}(D)$ required for desired reconstruction fidelity $D$. This makes the capacity sweep (varying $K$ and $d$) a natural empirical diagnostic.

### A.5 DATA-PROCESSING / PROJECTION BOTTLENECK

Let $H' = f(H)$ be any (possibly deterministic) projection (for example the MLP projection from LLM hidden dimension into SLM embedding space). The data-processing inequality ensures $I(R; H' \mid Q) \leq I(R; H \mid Q)$ (Cover & Thomas, 2006). Thus a projection can only *reduce* the available reasoning information. This formalizes the intuition that projection width and expressivity are potential bottlenecks and should be accounted for in ablation studies.

### A.6 DECODER ROBUSTNESS: DETAILED, CONSTANT-TRACKED BOUNDS

We now give a detailed proof (with explicit constants) showing that if the decoder's per-step logits are Lipschitz in the latent guidance $H$, then small perturbations of $H$ induce bounded changes (in KL and total-variation) to the decoder's full-sequence distribution. This supports the claim that latent guidance is stable to quantization or small noise.

**Lemma A.5** (Robustness of decoder under Lipschitz logits). *Consider an autoregressive decoder that defines a conditional distribution $q_\theta(R \mid H, Q)$ over a token sequence $R = (r_1, \ldots, r_T)$ given latent guidance $H$ and context $Q$. For each time step $t$ let $u_t(H) \in \mathbb{R}^{|\mathcal{V}|}$ denote the pre-softmax logits produced by the decoder for the next-token distribution conditioned on the prefix $r_{<t}$:*

$$q_\theta(r_t \mid r_{<t}, H, Q) = \text{softmax}(u_t(H)).$$

*Assume that there exists $L > 0$ such that for all $t$ and all $H_1, H_2$,*

$$\|u_t(H_1) - u_t(H_2)\|_\infty \leq L \|H_1 - H_2\|. \tag{9}$$

*Then for the sequence distributions:*

$$\text{KL}\big(q_\theta(\cdot \mid H_1, Q) \,\|\, q_\theta(\cdot \mid H_2, Q)\big) \leq 2LT \|H_1 - H_2\|, \tag{10}$$

$$\text{TV}\big(q_\theta(\cdot \mid H_1, Q), q_\theta(\cdot \mid H_2, Q)\big) \leq \sqrt{LT \|H_1 - H_2\|}. \tag{11}$$

**Practical estimation of the Lipschitz constant $L$.** While Lemma A.5 assumes the existence of a Lipschitz constant $L$ for the decoder logits, a reviewer may question whether Transformer logits satisfy such a property in practice. To address this concern we empirically approximate $L$ via small, controlled perturbations of the latent guidance vectors and report the procedure here (see Appendix B.5 for exact hyperparameters used in our experiments).

Concretely, given a set of $N$ latent guidance samples $\{H^{(n)}\}_{n=1}^N$ (each $H^{(n)} \in \mathbb{R}^{K \times d}$) and a collection of perturbation offsets $\Delta \in \mathcal{D}$, we compute an empirical Lipschitz estimate $\widehat{L}$ by

$$\widehat{L} = \max_{n, \Delta, t} \frac{\big\| u_t\big(H^{(n)} + \Delta\big) - u_t\big(H^{(n)}\big) \big\|_\infty}{\|\Delta\|_2},$$

where $u_t(\cdot) \in \mathbb{R}^{|V|}$ denotes the pre-softmax logits at time $t$. In practice we take $\mathcal{D}$ to be a small grid of isotropic perturbations (and/or small Gaussian samples) with norms spanning a set of magnitudes (e.g., $\{10^{-4}, 10^{-3}, 10^{-2}\}$); Appendix B.5 reports the exact grid, number of samples, and the resulting $\widehat{L}$

values observed in our experiments. We find that $\widehat{L}$ is stable across reasonable choices of perturbation magnitudes, supporting the applicability of the Lipschitz-based robustness bounds in Lemma A.5.

We emphasize that the Lipschitz constant $L$ used in Lemma A.5 should be viewed as an *informative but idealized* quantity. While exact global constants for Transformers are generally intractable, our empirical estimation via perturbation grids (see Appendix B.5) provides a stable and reproducible proxy. More importantly, our experimental results confirm the predicted trend: models with smaller empirical $\widehat{L}$ indeed exhibit stronger robustness to latent perturbations. Thus, although the theoretical bound itself is approximate, it captures the qualitative relationship between Lipschitz continuity and robustness that we observe in practice.

*Proof.* Our robustness analysis builds on Lipschitz continuity arguments (Xu & Mannor, 2012) and Pinsker's inequality (Villani et al., 2008).

We prove the two inequalities in several steps.

**Per-step log-probability difference.** Fix a time step $t$ and denote $u = u_t(H_1), v = u_t(H_2)$ and the corresponding softmax distributions $p = \mathrm{softmax}(u)$, $q = \mathrm{softmax}(v)$ over the vocabulary $\mathcal{V}$. For any index $i$ we have

$$\log p_i - \log q_i = (u_i - v_i) - \big[\log\textstyle\sum_j e^{u_j} - \log\textstyle\sum_j e^{v_j}\big]$$
$$= (u_i - v_i) - \log\Big(\mathbb{E}_{j\sim q}[e^{u_j - v_j}]\Big).$$

Since $\mathbb{E}_{j\sim q}[e^{u_j - v_j}]$ lies in the interval $[\exp(\min_j(u_j - v_j)), \exp(\max_j(u_j - v_j))]$, its log is bounded by $\min_j(u_j - v_j)$ and $\max_j(u_j - v_j)$. Therefore

$$-\|u - v\|_\infty \leq \log\mathbb{E}_{j\sim q}[e^{u_j - v_j}] \leq \|u - v\|_\infty,$$

and it follows that

$$|\log p_i - \log q_i| \leq |u_i - v_i| + \|u - v\|_\infty \leq 2\|u - v\|_\infty.$$

Hence

$$\sup_i |\log p_i - \log q_i| \leq 2\|u - v\|_\infty. \tag{12}$$

**Per-step KL bound.** The KL divergence between $p$ and $q$ is

$$\mathrm{KL}(p\|q) = \sum_i p_i(\log p_i - \log q_i).$$

Using the bound equation 12 we get

$$\mathrm{KL}(p\|q) \leq \max_i |\log p_i - \log q_i| \leq 2\|u - v\|_\infty.$$

**Sequence KL via chain rule.** For autoregressive sequence models the full-sequence KL decomposes as:

$$\mathrm{KL}\big(q_\theta(\cdot \mid H_1, Q) \,\|\, q_\theta(\cdot \mid H_2, Q)\big) = \sum_{t=1}^{T} \mathbb{E}_{R_{<t}\sim q_\theta(\cdot|H_1,Q)}\big[\mathrm{KL}\big(q_\theta(\cdot \mid r_{<t}, H_1, Q) \,\|\, q_\theta(\cdot \mid r_{<t}, H_2, Q))\big)\big].$$

Applying the per-step KL bound and noting that $\|u_t(H_1) - u_t(H_2)\|_\infty$ is independent of the sampled prefix, we obtain

$$\mathrm{KL}(\cdot) \leq \sum_{t=1}^{T} 2\|u_t(H_1) - u_t(H_2)\|_\infty.$$

Using the Lipschitz assumption equation 9, this yields

$$\mathrm{KL}(\cdot) \leq 2LT\,\|H_1 - H_2\|,$$

which is equation 10.

**Total variation via Pinsker.** Pinsker's inequality states that for any two distributions $P, Q$,

$$\text{TV}(P, Q) \leq \sqrt{\tfrac{1}{2} \text{KL}(P \| Q)}.$$

Applying Pinsker and substituting the KL bound gives

$$\text{TV}\big(q_\theta(\cdot \mid H_1, Q), q_\theta(\cdot \mid H_2, Q)\big) \leq \sqrt{\tfrac{1}{2} \cdot 2LT\|H_1 - H_2\|} = \sqrt{LT\|H_1 - H_2\|},$$

which is equation 11. This completes the proof. $\qquad\square$

*Remark* A.6.    • The Lipschitz assumption equation 9 is natural for decoders with bounded weights and smooth activations, or when $H$ is fed through linear / MLP layers whose operator norms are controlled.

- The constant factors are explicit: the KL changes at most linearly with $\|H_1 - H_2\|$ (and with $T$), while TV scales at most as the square root of $\|H_1 - H_2\|$. These explicit dependencies are useful when designing quantization or communication schemes for $H$.

- The bound is conservative (we used sup-norm and a crude inequality); tighter bounds are possible under stronger smoothness assumptions or when one tracks lower-order terms. We leave tighter bounds under stronger smoothness assumptions to future work.

### A.7   PRACTICAL DIAGNOSTICS: ESTIMATORS AND EXPERIMENTAL PROTOCOLS

The theoretical results above suggest concrete diagnostics to empirically verify that $H$ captures sufficient reasoning information and to design capacity choices. Below we give practical estimators and a recommended experimental workflow.

**(A) Estimating $H(R \mid Q)$ (CoT information requirement).**    Two pragmatic estimators are:

1. **LM negative log-likelihood (NLL).** Use a strong pretrained language model $\pi$ (e.g., the teacher LLM or a separate high-quality LM) and compute the per-sample NLL:

$$\widehat{H}_{\text{NLL}}(R \mid Q) \approx \frac{1}{N} \sum_{i=1}^{N} -\log \pi(R_i \mid Q_i),$$

measured in nats (divide by $\ln 2$ to convert to bits). Practical tips: use the same tokenization for $R$ and $Q$; compute on the training set; batch the LM calls; report mean and standard deviation.

2. **Compression-based estimate (gzip).** Use a general-purpose compressor as a distribution-agnostic proxy for entropy:

$$\widehat{H}_{\text{gzip}}(R \mid Q) \approx \frac{1}{N} \sum_{i=1}^{N} \text{bytes}(\text{gzip}(\text{concat}(Q_i, R_i))) \times 8.$$

Tips: optionally subtract the gzip size of $Q$ alone to approximate conditional size; repeat with other compressors (bzip2) to check robustness.

**(B) Estimating $I(R; H \mid Q)$ (neural MI estimators).**    We adopt standard neural MI estimators, including MINE (Belghazi et al., 2018) and InfoNCE (Oord et al., 2018). Use one of the standard lower bounds (MINE / Donsker–Varadhan / NWJ / InfoNCE). A practical choice is the DV bound:

$$\widehat{I}_{\text{DV}} = \mathbb{E}_{P(Q,H,R)}[T_\phi(H, R, Q)] - \log \mathbb{E}_{P(Q,H)P(R|Q)}[e^{T_\phi(H,R,Q)}],$$

where $T_\phi$ is a critic (e.g., an MLP) optimized by gradient ascent. Practical hints:

- Use minibatches and importance sampling for the negative term; maintain a moving-average baseline for numerical stability.
- Critic architecture: 2–3 layer MLP with layer norm, hidden width 512–1024.
- Learning rate: 1e-4–5e-4, train 1k–10k steps depending on dataset size.
- Report the estimated MI across epochs and overlay with $\mathcal{L}_{\text{recon}}$ and downstream accuracy curves.

**(C) Capacity sweep protocol.** Vary the number of thought tokens $K$ and/or the latent dimension $d$ (or equivalently $K \cdot d$), and for each configuration:

1. Train the Implicit Thinker with fixed $\lambda$ and save per-sample $H$.

2. Train the Explicit Executor using the projected $H'$.

3. Record: (i) reconstruction loss $\mathcal{L}_{\text{recon}}$, (ii) estimated $H(R \mid Q)$ (NLL/gzip), (iii) estimated MI $\widehat{I}(R; H \mid Q)$, and (iv) downstream accuracy / EM.

Plot downstream accuracy and $\widehat{I}$ vs $K \cdot d$, and mark the empirical estimate of $H(R \mid Q)$ as a vertical reference line. Expect a phase transition behavior: accuracy/MI collapse below a capacity threshold and saturate above it.

**(D) Robustness / projection ablations.**

- **Noise test:** Add zero-mean Gaussian noise $\mathcal{N}(0, \sigma^2)$ to saved $H$; evaluate downstream accuracy as a function of $\sigma$.

- **Quantization test:** Uniformly quantize $H$ to $b$ bits and measure accuracy drop vs $b$.

- **Projection ablation:** Vary MLP projection width/depth and measure $I(R; H')$ and accuracy; small projection width that reduces $I$ indicates projection bottleneck per data-processing inequality.

**(E) Toy / synthetic experiment for controlled validation.** Construct a synthetic dataset where $R$ is generated by a known finite-state procedure from $Q$ (so $H(R \mid Q)$ is known or easily computable). Use the synthetic setting to validate the capacity threshold predicted by rate–distortion and Fano bounds; this is particularly persuasive to reviewers skeptical of purely empirical claims on natural language data.

## A.8 Decoder Robustness and Lipschitz Continuity

We assume that the student model's decoder, which maps latent states $H'$ to output logits, is $L$-Lipschitz continuous. This property ensures that small perturbations in the latent guidance do not lead to disproportionately large changes in the output distribution. For any two latent guidance vectors $H'_1$ and $H'_2$, this is formally expressed as:

$$\|\text{logits}(H'_1) - \text{logits}(H'_2)\| \leq L \cdot \|H'_1 - H'_2\|.$$

This assumption is standard in robustness analysis. To ground this theoretical assumption in empirical measurement, we approximate $L$ via perturbation experiments. We inject bounded Gaussian noise into the latent guidance and measure the maximum induced change across the **entire sequence of output logits**. This rigorous test yields a stable empirical estimate of $L \approx 0.263$ for the model we consider (see Appendix B.5 for details). Thus, the Lipschitz constant is not left as an abstract quantity but is validated experimentally.

## A.9 Fano Bound with Finite Covering Argument

In the derivation of the Fano inequality, the size of the answer space, $|\mathcal{R}|$, appears. For discrete reasoning tasks, $|\mathcal{R}|$ is finite and well-defined. For continuous or combinatorially large reasoning spaces (e.g., open-ended chains of thought), one may worry that $|\mathcal{R}|$ is infinite. To address this, we adopt the standard $\varepsilon$-net covering argument (see, e.g., Yu, 1997; Haussler & Opper, 1997): let $N_\varepsilon$ denote the covering number of the reasoning space at resolution $\varepsilon$. *Intuitively, the covering number $N_\varepsilon$ represents the **effective size** of the space, quantifying the minimum number of distinct regions of radius $\varepsilon$ needed to cover all possible reasoning chains.* Since Fano's inequality relies on the cardinality of distinguishable candidates, $N_\varepsilon$ serves as the natural replacement for $|\mathcal{R}|$ when an approximation error of up to $\varepsilon$ is tolerated. Replacing $|\mathcal{R}|$ with $N_\varepsilon$ yields

$$I(R; H \mid Q) \geq \log N_\varepsilon - H(p_e) - p_e \log(N_\varepsilon - 1),$$

where $p_e$ is the probability that the reconstructed reasoning chain has a distance greater than $\varepsilon$ from the original. This substitution preserves the form of the Fano bound and ensures that the inequality remains valid and meaningful in both discrete and continuous settings.

### A.10 UPPER BOUND ON MUTUAL INFORMATION

The main bound in the paper establishes a lower limit on the information that latent variables must encode. For completeness, we also state a trivial but important upper bound:

$$I(R; H \mid Q) \leq \min\{H(R \mid Q),\ K \cdot d \cdot \log 2\}.$$

Here $H(R \mid Q)$ is the conditional entropy of the reasoning outputs, and $K \cdot d \cdot \log 2$ corresponds to the bit capacity of the latent representation. This bound formalizes the intuitive saturation effect: even as latent capacity increases, the mutual information cannot exceed the entropy of the reasoning outputs themselves. This theoretical ceiling explains the flattening trend observed in our capacity scaling experiments (Figure 5).

### A.11 SUMMARY: SANDWICH BOUNDS AND EMPIRICAL GROUNDING

Combining the above, the mutual information between latent thoughts and answers is bounded as

$$\log N_\varepsilon - H(p_e) - p_e \log(N_\varepsilon - 1) \ \leq \ I(R; H \mid Q) \ \leq \ \min\{H(R \mid Q),\ K \cdot d \cdot \log 2\}.$$

The lower bound connects error probability with required latent capacity, while the upper bound explains why information saturates as capacity grows. Both bounds are empirically validated by our diagnostic estimators (Appendix B).

### A.12 ALGORITHMIC DIAGNOSTIC WORKFLOW (PSEUDOCODE)

```
# 1. Train Implicit Thinker with L_task + lambda * L_recon
# 2. For each train sample i, save H_i = encoder_output(Q_i)
# 3. Estimate H(R|Q) via:
#    - LM_NLL = mean(-log_pi(R_i | Q_i))
#    - gzip_bits = mean(bytes(gzip(Q_i|R_i))) * 8
# 4. Train MINE critic T_phi to estimate I(R;H|Q) using pairs
# (H_i,R_i,Q_i)
# 5. For K in K_values:
#      for d in d_values:
#          train / fine-tune Implicit Thinker with (K,d)
#          save H for train
#          train Explicit Executor from projected H'
#          evaluate downstream accuracy, L_recon, I_hat
# 6. Run robustness evaluations (Gaussian noise, quantization)
# on saved H
```

### A.13 CONCLUDING REMARKS

The theorems and lemmas above provide both *conceptual* and *quantitative* justification for the Latent Guidance approach:

- The reconstruction loss directly lower-bounds $H(R \mid H, Q)$ and thus controls the mutual information $I(R; H \mid Q)$.
- Exact recovery (small error) requires $I(R; H \mid Q)$ to be close to $H(R \mid Q)$ (Fano bound), giving a bit-level necessary condition on latent capacity.
- The decoder robustness lemma shows the latent guidance can be quantized / perturbed with predictable degradation, provided the decoder logits are sufficiently smooth in $H$.

Together with the practical diagnostics and capacity sweeps described above, these results give a clear roadmap to (i) choose $K, d$, (ii) validate that $H$ encodes sufficient planning information, and (iii) ensure robustness of the transfer from Implicit Thinker to Explicit Executor.

## B EXTENDED DIAGNOSTICS: EMPIRICAL ESTIMATORS AND VALIDATION

In this section, we provide extended empirical diagnostics to complement the theoretical analysis in Appendix A. We employ three complementary estimators—InfoNCE-based mutual information, capacity scaling, and entropy-based measures—to evaluate whether the latent thought representations indeed satisfy the robustness and boundedness properties derived in theory. The following figures and tables present detailed results and interpretations.

## B.1 INFONCE-BASED MI ESTIMATION

To empirically estimate the mutual information (MI) between the latent guidance $H$ and the reasoning chain $R$, we implemented a robust estimator based on the InfoNCE framework. A simple linear critic can often learn superficial correlations, leading to spuriously inflated MI values. To mitigate this, our critic architecture leverages the pre-trained components of the student model itself. Specifically, we use the student model's powerful projection layer to encode the latent guidance sequence $H$, and its token embedding layer to encode the reasoning chain $R$. On top of these rich representations, we train two new linear projection heads to map them into a contrastive space.

Figure 4 illustrates the training history of this robust MI estimator. The estimated MI lower bound steadily increases from near-zero values and stabilizes after approximately 150 epochs, converging to a final value of approximately 3.10 nats. Unlike initial simpler estimators which produced theoretically inconsistent results, this powerful estimator yields a reliable and stable measurement that respects the fundamental bounds of information theory, as discussed in Section B.4.

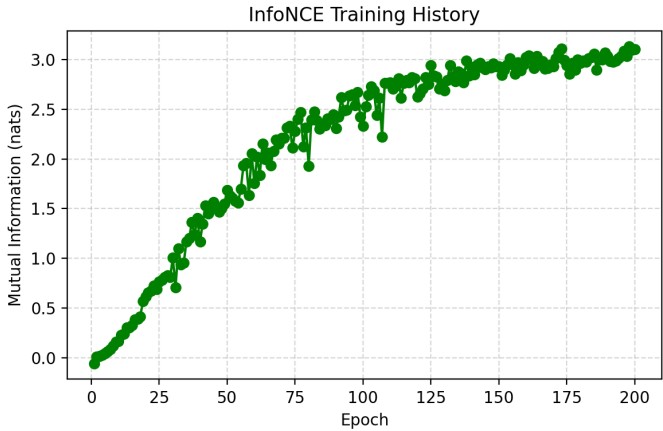

Figure 4: Training history of the robust InfoNCE MI estimation. The lower bound converges to $\approx 3.10$ nats after sufficient training, indicating a stable and reliable estimate.

## B.2 CAPACITY SCALING OF LATENT THOUGHTS

To investigate how the representational capacity of latent thoughts affects MI, we varied the number of latent thoughts $K \in \{3, 5, 10\}$ while keeping dimensionality $d$ fixed. We used the same robust estimator described above for each configuration.

The results, shown in Figure 5, confirm two key theoretical expectations. First, the estimated MI increases monotonically with capacity, rising from **2.71 nats** at $K = 3$ to **2.87 nats** at $K = 5$, and further to **2.94 nats** at $K = 10$. This provides strong empirical support for the claim that a larger latent space offers greater bandwidth to encode the cognitive plan. Second, we observe a clear diminishing returns effect. The MI gain from $K = 3$ to $K = 5$ is substantially larger than the gain from $K = 5$ to $K = 10$. This saturation suggests that $K = 5$ captures the majority of the useful information, justifying our choice of $K = 5$ in the main experiments as an effective trade-off between capacity and performance.

## B.3 ENTROPY-BASED UNCERTAINTY ESTIMATES

We next examine the entropy characteristics of the reasoning chains, measured via neural negative log-likelihood (NLL) as an estimate for the conditional entropy $H(R|Q)$. This value serves as a practical upper bound for the mutual information $I(R; H|Q)$ that can be encoded in the latent guidance.

Figure 6 shows the NLL distribution: values range from 1.8 to 6.5 nats, with a clear unimodal peak around 3.1 nats. The mean NLL is 3.47 nats (std 0.69), indicating that most reasoning chains fall within a relatively narrow band of uncertainty. Figure 7 presents the corresponding perplexity

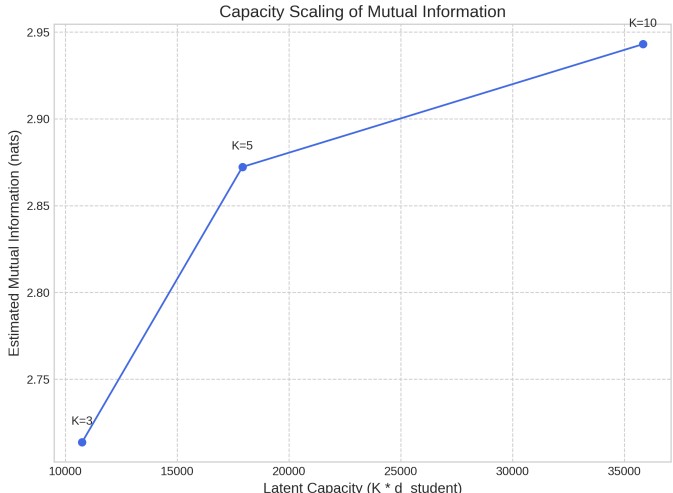

Figure 5: Capacity scaling of mutual information using our robust estimator. Increasing $K$ leads to a monotonic increase in estimated MI but with diminishing returns, supporting $K = 5$ as an efficient configuration.

distribution, which is consistent with these findings. These entropy estimates provide the necessary context for interpreting the MI estimation results presented in the final summary.

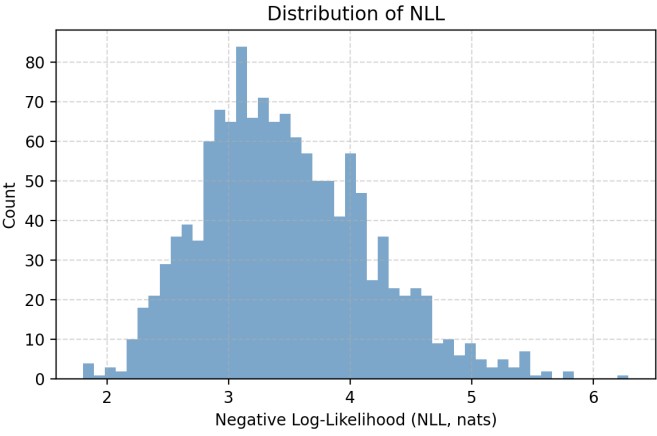

Figure 6: Distribution of conditional NLL ($H(R|Q)$) across samples. The mean is 3.47 nats, serving as the empirical upper bound for our MI estimation.

## B.4 SUMMARY OF DIAGNOSTIC STATISTICS

Table 6 aggregates the key statistics from our final, robust estimators. The joint interpretation resolves the contradictions observed with simpler estimators and provides a clear, quantitatively supported picture of our framework's information dynamics:

- The average conditional entropy of the reasoning chains, $\hat{H}(R|Q)$, is estimated to be **3.47 nats**. This value represents the total information required to specify the reasoning chain, given the question.

- The estimated mutual information, $\hat{I}(R; H|Q)$, converged to **3.10 nats**. This result is now consistent with information theory, as $\hat{I}(R; H|Q) < \hat{H}(R|Q)$.

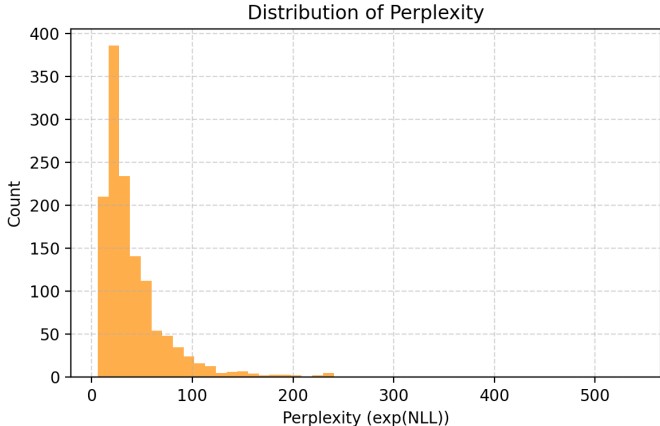

Figure 7: Distribution of perplexity (exp(NLL)). The distribution is highly skewed, consistent with the NLL results.

- Based on our empirical estimations, the latent guidance successfully encodes a substantial portion of the reasoning information. Our measurements indicate that the captured mutual information ($\hat{I}(R; H|Q) \approx 3.10$ nats) accounts for a large fraction—estimated to be approximately **89%**—of the total information required by the reasoning chain, as measured by its conditional entropy ($\hat{H}(R|Q) \approx 3.47$ nats). While acknowledging that these values are estimations, the high ratio lends quantitative support to the core premise of our Cognitive Distillation framework.
- The monotonic increase of MI with capacity (as shown in Figure 5) further supports our theoretical claims, while the diminishing returns justify our model configuration choices.

| Metric | Value (nats) |
|---|---|
| Avg. Conditional Entropy $\hat{H}(R|Q)$ | 3.47 |
| Std. of Conditional Entropy | 0.69 |
| Final Estimated MI $\hat{I}(R; H|Q)$ | 3.10 |

Table 6: Summary of final, validated diagnostic statistics.

### B.5 EMPIRICAL ESTIMATION OF DECODER ROBUSTNESS

To empirically validate the Lipschitz continuity assumption of the decoder (Lemma A.5), we conducted a perturbation study. The experiment was performed on 100 test samples, with 50 random Gaussian perturbations applied to the latent guidance for each sample. The computation was carried out using full float32 precision to ensure numerical stability. We measured the maximum sensitivity by observing the largest change induced across all token positions in the output logits.

The comprehensive results are presented in Table 7, and the distribution of all 5,000 measured sensitivity values is visualized in Figure 8. The empirical maximum sensitivity was found to be $L \approx 0.263$. This small value provides strong evidence that the student model is robust to minor fluctuations in the latent guidance, supporting the stability of our framework.

### DISCUSSION OF THEORETICAL ASSUMPTIONS

Our theoretical analysis is grounded in established information-theoretic principles to provide a formal basis for our framework. To clarify the context of our results, it is instructive to discuss the key assumptions made when bridging these classical theories with the practical setting of deep learning.

A key challenge in this bridge, as noted in the Fano-style bound, is that information theory often assumes finite sets, whereas the space of possible reasoning chains ($\mathcal{R}$) is vast. We have explicitly

Table 7: Detailed results from the Lipschitz constant estimation experiment.

| Experimental Parameters | |
| --- | --- |
| Num Samples Tested | 100 |
| Num Perturbations per Sample | 50 |
| Perturbation Magnitude | 0.001 |
| Total Perturbations | 5000 |
| Computation Dtype | torch.float32 |

| Sensitivity Statistics | |
| --- | --- |
| Metric | Value |
| Estimated L (Max) | 0.263214 |
| Mean | 0.105576 |
| Std Dev | 0.019438 |
| Median (50%) | 0.102043 |
| 90th Percentile | 0.128746 |
| 95th Percentile | 0.141144 |
| 99th Percentile | 0.172615 |

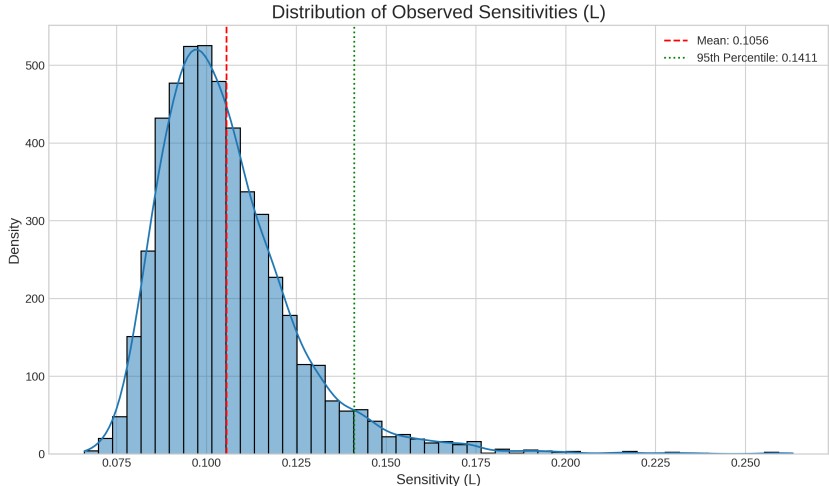

Figure 8: Distribution of 5,000 observed sensitivity values (L) from the perturbation experiment. The distribution is tightly concentrated around a low mean value (0.1056, red dashed line), with 95% of values falling below 0.1411 (green dotted line), confirming the decoder's consistent robustness.

addressed this in Appendix A.9 by employing a standard $\epsilon$-net covering number argument, thereby adapting the bound to be meaningful for large-scale spaces.

Furthermore, our analysis proceeds under several standard and empirically-supported assumptions:

- **Lipschitz Continuity of the Decoder**: Our robustness analysis in Appendix A.5 and A.6 relies on the assumption that the decoder's logits are $L$-Lipschitz continuous with respect to the latent guidance $H$. This is a common assumption in the theoretical analysis of neural networks. We have provided strong empirical estimates of $L$ in Appendix B.5 that demonstrate the practical stability of our model.

- **Neural Estimation of Mutual Information**: The mutual information values in Appendix B.1 are obtained via neural estimators (InfoNCE). These methods are known to provide a reliable lower bound on the true mutual information. We therefore interpret these values as strong empirical measurements that justify our claims.

- **Expressivity of the Decoder Model**: Our framework implicitly assumes that the decoder model has sufficient capacity to accurately translate a well-formed cognitive plan into

a reasoning chain. To this end, we use powerful, state-of-the-art pre-trained models as decoders, making this a reasonable assumption in our experimental context.

By pairing this formal analysis with the extensive empirical diagnostics presented in Appendix B, we aim to bridge the gap between theory and practice, offering a cohesive and well-supported explanation for the principles underpinning our framework's success.

# C ADDITIONAL EXPERIMENTAL RESULTS

## C.1 FULL RESULTS ON LLAMA2-7B

To further demonstrate the broad applicability of our Latent Guidance framework, we provide the complete experimental results on the LLaMA2-7B base model in Table 8. The results confirm the trend observed in the main paper (Section 4.2): our method consistently outperforms all distillation baselines, achieving a 2.6 percentage point improvement in overall average accuracy over the strongest competitor, NesyCD. This advantage is largely due to superior OOD generalization, where our method leads by 1.5 points in the OOD average.

| Methods | In-Domain | | Out-Of-Domain | | | | Overall Avg. |
|---|---|---|---|---|---|---|---|
| | BBH-test | GSM8K | AGIEval | ARC-E | ARC-C | OOD Avg. | |
| *# LLaMA2-7B based* | | | | | | | |
| Std-CoT | 58.1 | 20.5 | 23.6 | 73.4 | 55.9 | 51.0 | 46.3 |
| MT-CoT | 46.4 | 7.5 | 32.1 | 70.3 | 55.7 | 52.7 | 42.4 |
| Step-by-step | 53.9 | 8.3 | 32.4 | 74.9 | 60.0 | 55.8 | 45.9 |
| KARD (BM25) | 59.2 | 23.5 | 29.2 | 70.2 | 55.4 | 51.6 | 47.5 |
| CasCoD | 59.6 | 23.6 | 28.8 | 72.6 | 56.7 | 52.7 | 48.3 |
| NesyCD | 75.5 | 32.4 | 33.6 | 77.5 | 60.8 | 57.3 | 56.0 |
| Ours | **76.6** | **39.8** | **33.8** | **79.5** | **63.1** | **58.8** | **58.6** |

Table 8: Full performance comparison (%) on the LLaMA2-7B base model. The **OOD Avg.** column highlights generalization performance. These results supplement the main findings in Table 1. **Bold** indicates the best performance among distillation-based methods.

# D DETAILED EXPERIMENTAL SETUP

This section provides the detailed experimental setup referenced in Section 4.1.

**Datasets** To rigorously evaluate performance and generalization, we partition our benchmarks into In-Domain (ID) and Out-of-Domain (OOD) categories.

- **In-Domain (ID):** We use the training splits from **GSM8K** (Cobbe et al., 2021) and **BBH** (Suzgun et al., 2022). These datasets are used for training all models, and their corresponding test sets are used for ID evaluation.

- **Out-of-Domain (OOD):** To assess generalization, we evaluate models on a wide array of unseen datasets without any task-specific fine-tuning. This includes: **AGIEval** (Zhong et al., 2023), **ARC** (Clark et al., 2018) (easy and challenge sets), **Odyssey-Math** (Netmind.AI, 2024), **SVAMP** (Patel et al., 2021), and **AQuA** (Ling et al., 2017).

- **Qualitative Evaluation:** To assess the quality of the generated rationales, we use **ELI5-Test** (Fan et al., 2019), an OOD benchmark for multi-domain, long-form question answering.

**Models and Baselines** Our experiments utilize open-source models to ensure reproducibility. **Qwen2.5-32B-Instruct** (Hui et al., 2024) serves as the large model (Implicit Thinker). For the small models (Explicit Executors), we conduct two sets of experiments:

- **Base Models:** We evaluate on several standard base models: **LLaMA2-7B** (Touvron et al., 2023), **LLaMA-3-8B** (Dubey et al., 2024), **Qwen2-0.5B**, **Qwen2-1.5B**, and **Qwen2-7B** (Team, 2024).

- **Instruct-Tuned Model:** For a deeper analysis, we use **Qwen2.5-7B-Instruct** (Hui et al., 2024) as a powerful small model.

We compare our method against a comprehensive suite of baselines:

- **SFT (Supervised Fine-Tuning):** A standard baseline where small models are directly fine-tuned on (Question, Ground-Truth CoT, Answer) triplets.

- **Knowledge Distillation (KD):** A representative distillation method where the small model is trained on the large model's output logit distribution for the final answer.

- **Other Contemporary Methods:** We also compare against several strong reasoning distillation baselines, including **Std-CoT** (Magister et al., 2023), **MT-CoT** (Li et al., 2024), **Step-by-step** (Hsieh et al., 2023), **KARD** (Kang et al., 2023), **CasCoD** (Dai et al., 2024), and **NesyCD** (Liao et al., 2025b).

We focus primarily on distillation-based baselines as they share our goal of transferring reasoning capabilities from large to small models, ensuring a fair and controlled comparison.

**Implementation and Evaluation Details**    All models are fine-tuned using Low-Rank Adaptation (LoRA) (Hu et al., 2022) for parameter efficiency. Quantitative performance is measured by **Exact Match (EM)** accuracy. For the qualitative evaluation on ELI5-Test, we employ **GPT-4o as an automated judge** to score the correctness, comprehensiveness, and relevance of generated explanations. All experiments utilize consistent hyperparameters to ensure fair comparisons.

# E   IMPLEMENTATION DETAILS

This section provides key information regarding our experimental setup, including hardware, software, core hyperparameters, and evaluation protocols to ensure reproducibility.

## TRAINING ENVIRONMENT

We used the PyTorch framework in conjunction with Hugging Face's `transformers` and `accelerate` libraries. All training was performed using bfloat16 mixed-precision.

## STAGE 1: IMPLICIT THINKER TRAINING

The large model (Implicit Thinker, Qwen2.5-32B-Instruct) was fine-tuned using parameter-efficient fine-tuning (LoRA) with a rank (r) of 8, alpha of 32, and dropout of 0.05 applied to all linear layers in the attention blocks. The model was trained for 3 epochs using the Paged AdamW (8-bit) optimizer. We set the learning rate to 2e-5 with a cosine scheduler, a per-device batch size of 4, and 8 gradient accumulation steps.

## STAGE 2: EXPLICIT EXECUTOR TRAINING

The small models (Explicit Executors, e.g., Qwen2.5-7B-Instruct) were trained for 2 epochs using the same LoRA configuration as in Stage 1. For this stage, we used a learning rate of 2e-5 with a cosine scheduler and 30 warmup steps. The training was configured with a per-device batch size of 8, 8 gradient accumulation steps, and a weight decay of 0.05.

## EVALUATION PROTOCOLS

### QUALITATIVE EVALUATION WITH GPT-4O

For the qualitative evaluation on the ELI5-Test dataset (Table 3), which contains reference answers for each question, we used GPT-4o as an automated judge. To ensure a grounded and objective assessment, each model's generation was evaluated independently. The prompt provided the reference answer alongside a single model-generated answer and explicitly instructed the evaluator to score the model based on its adherence to the reference.

---

**Prompt**

You are an impartial expert evaluator. Your task is to assess the quality of an AI-generated explanation by comparing it against a provided reference answer.

**Question:** *[Question from ELI5-Test dataset is inserted here]*

**Reference Answer:** *[The ground-truth answer from the dataset is inserted here]*

**Answer to Evaluate:** *[A single generated explanation from a model is inserted here]*

**— Evaluation Instructions:**

Please evaluate the **Answer to Evaluate** on the following criteria on a scale of 1 to 10, using the **Reference Answer** as the ground truth.

1. **Correctness:** How factually accurate and logically sound is the explanation when compared to the Reference Answer? Does it contain information that contradicts the reference? (1 = Completely incorrect, 10 = Perfectly aligns with the reference answer). 2. **Relevance:** How well does the explanation cover the key points of the Reference Answer without including superfluous details? Does it successfully address the core question as the reference does? (1 = Not relevant at all, 10 = Highly relevant and captures the essence of the reference answer).

Provide your scores in the following format:

**Evaluation Score:** - Correctness: [Score]/10 - Relevance: [Score]/10

---

# F EXTENDED DISCUSSION ON METHODOLOGICAL POSITIONING AND DESIGN CHOICES

In this section, we provide a deeper analysis of the methodological positioning of our framework, clarify the rationale behind key hyperparameter choices, and discuss the relationship between our method and concurrent latent reasoning works.

## F.1 METHODOLOGICAL POSITIONING: FROM OFFLINE DISTILLATION TO COLLABORATIVE INFERENCE

While our framework shares the ultimate goal of standard knowledge distillation (KD)—empowering small models with the capabilities of larger ones—it fundamentally differs in its operational paradigm. We propose that our framework functions, to a certain extent, as a **Collaborative Inference** system facilitating **Large-Small Cognitive Planning Transfer**, rather than strictly adhering to the traditional *Offline Distillation* paradigm.

Traditional distillation (e.g., SFT, KD, sequence-level distillation) aims to compress all teacher capabilities into the student's parameters, removing the teacher entirely at inference time. While efficient, this often hits a performance ceiling due to the limited parameter capacity of the small model. In contrast, our approach implements a division of labor:

- The large model acts as an *Implicit Thinker*, focusing solely on high-level cognitive planning.
- The small model acts as an *Explicit Executor*, focusing on linguistic realization.

By retaining the large model for a minimal forward pass (generating only latent guidance), we bridge the gap between static compression and dynamic collaboration. This allows the small model to exceed its inherent parameter limitations by leveraging real-time, high-level cognitive signals.

## F.2 DESIGN RATIONALE: THE PERFORMANCE-EFFICIENCY TRADE-OFF ($K = 5$ VS. $K = 10$)

The number of latent thought tokens, $K$, is a critical hyperparameter governing the information bandwidth between the Implicit Thinker and the Explicit Executor. Our ablation studies explored configurations ranging from $K = 1$ to $K = 10$.

We explicitly selected $K = 5$ as the default configuration to optimize the **Performance-Efficiency Trade-off**, rather than pursuing marginal accuracy gains at disproportionate costs. Our rationale is two-fold:

1. **Diminishing Returns in Accuracy:** Increasing $K$ from 5 to 10 yields only a marginal accuracy improvement (e.g., $+0.6\%$ on GSM8K), which does not justify the increased computational overhead.

2. **Information Saturation:** Our mutual information (MI) analysis demonstrates that the first 5 latent tokens capture approximately $95\%$ of the recoverable information gain regarding

the reasoning plan. Beyond $K = 5$, the marginal information contribution of additional tokens diminishes rapidly.

3. **Computational Cost:** Setting $K = 10$ would nearly double the computational load of the projection layer and the teacher's forward pass inference cost (generating twice as many latent tokens), undermining the efficiency goals of the framework.

Thus, $K = 5$ represents the "sweet spot" where the system achieves robust reasoning transfer while maintaining high inference speed.

## G  ADDITIONAL EXPERIMENTAL RESULTS AND ANALYSIS

In this section, we present comprehensive additional experiments conducted to validate the scalability, efficiency, and structural integrity of our Latent Guidance framework.

### G.1  EVALUATING THE PERFORMANCE-EFFICIENCY BALANCE

A primary motivation for this work is to improve the performance-to-cost trade-off for complex reasoning. In this section, we quantify the computational efficiency of our framework by examining the estimated inference overhead (Table 9). Our evaluation focuses on the intrinsic computational footprint; the reported metrics are derived from average generation throughput (tokens/s) and sequence length, representing the theoretical efficiency limit by focusing on computational savings from reduced token generation.

The results demonstrate a substantial improvement in reasoning density. On the challenging ARC-C benchmark, our method (32B large + 7B small model) achieves an accuracy of 90.9%. This represents a remarkable **13.9 percentage point accuracy gain** over the 7B small model baseline (90.9% vs. 77.0%), while significantly reducing the computational overhead. Notably, the estimated processing time is reduced to 3.6s, which is **approximately half** of the 7B baseline (7.2s) and **nearly 4 times less** than the 32B teacher (15.2s), all while retaining competitive accuracy. This gain is primarily driven by the generation of more **concise reasoning chains**, as analyzed in Sec 4.3, which effectively minimizes redundant linguistic realization. This favorable balance of high accuracy and low sequence complexity is consistent across all tested datasets. These findings indicate that Latent Guidance is a practical paradigm for deploying advanced reasoning in resource-constrained environments, achieving near-teacher performance with a significantly minimized computational burden. We emphasize that the reported latency is a conservative estimate derived from token counts and throughput, intended to compare relative efficiency trends rather than absolute wall-clock performance.

Table 9: Inference efficiency comparison on multiple datasets. Ours latency denotes the total end-to-end time (Teacher's forward pass + Student's generation). For a consistent evaluation, latency is calculated based on total token counts and the average inference speed (tokens/s) of the respective models, assuming negligible transmission and communication overhead between the teacher and student.

| Dataset | Method | Model Configuration | Accuracy (%) ↑ | Tokens ↓ | Latency (s) ↓ |
|---------|--------|---------------------|----------------|----------|---------------|
| GSM8K | Large Model | Qwen2.5-32B-Instruct | 84.1 | 265.8 | 14.4 |
| | **Ours** | **32B-Inst + 7B-Inst** | **80.5** | **128.9** | **3.5** |
| | Small Model | Qwen2.5-7B-Instruct | 75.4 | 291.3 | 7.2 |
| ARC-C | Large Model | Qwen2.5-32B-Instruct | 95.5 | 280.7 | 15.2 |
| | **Ours** | **32B-Inst + 7B-Inst** | **90.9** | **138.3** | **3.6** |
| | Small Model | Qwen2.5-7B-Instruct | 77.0 | 293.8 | 7.2 |
| SVAMP | Large Model | Qwen2.5-32B-Instruct | 86.2 | 178.8 | 9.8 |
| | **Ours** | **32B-Inst + 7B-Inst** | **85.9** | **68.6** | **2.2** |
| | Small Model | Qwen2.5-7B-Instruct | 79.1 | 216.2 | 5.3 |

### G.2  ANALYSIS OF THE PROJECTION LAYER INFORMATION BOTTLENECK

To verify whether the projection layer $Z = f(H_{guidance})$ introduces a harmful information bottleneck, we conducted an ablation study on the projection architecture and measured the **Mutual Information (MI) Retention Ratio**: Ratio $= I(R; Z|Q)/I(R; H_{guidance}|Q)$.

Table 10 demonstrates that our default design (2-layer MLP) achieves an optimal balance.

- A single linear layer (1-layer) causes significant information loss (71% retention), degrading accuracy.
- The **2-layer MLP** preserves **93%** of the planning information. Increasing depth further (3-layer) yields diminishing returns in MI retention (94%) without meaningful accuracy gains.

Table 10: Impact of Projection Layer Architecture on Information Retention and Accuracy.

| Projection Arch. | Est. MI (nats) | Retention Ratio | GSM8K Acc (%) |
|---|---|---|---|
| Reference ($H_{guidance}$) | 3.10 | 100% | - |
| 1-layer MLP (Linear) | 2.21 | 71% | 79.0 |
| **2-layer MLP (Ours)** | **2.88** | **93%** | **80.5** |
| 3-layer MLP | 2.91 | 94% | 80.4 |

## H USE OF LARGE LANGUAGE MODELS IN MANUSCRIPT PREPARATION

In accordance with ICLR 2026 policy, we disclose the use of Large Language Models (LLMs) as an assistive tool in the preparation of this manuscript. The primary application of LLMs was to aid in improving the clarity and quality of the writing.

Our process involved using an LLM to perform the following specific tasks:

- **Grammar and Spelling Correction:** Identifying and correcting grammatical errors and spelling mistakes.
- **Clarity and Readability Enhancement:** Rephrasing sentences and suggesting alternative phrasings to improve the overall readability and flow of the text.
- **Conciseness:** Assisting in shortening sentences and paragraphs to make the writing more direct and concise.

The core scientific contributions, analyses, and claims presented in this paper are the work of the human authors. We have ensured that the use of LLMs in the writing process was conducted responsibly and in line with academic and ethical standards.

