# OpenReview forum: "Latent-Guided Reasoning: Empowering Small LLMs with Large-Model Thinking"
_ICLR.cc/2026/Conference — ICLR 2026 Poster_

### Official Review · Reviewer_bKFV · 2025-10-29

**Soundness:** 2
**Presentation:** 3
**Contribution:** 2
**Rating:** 2
**Confidence:** 2

**Summary:**

A new method is introduced to use latent reasoning from a large model (implicit thinker) to guide a smaller model to generate conventional CoT conditional on the latents, in order to improve the smaller model (which is trained under the inference setting).

**Strengths:**

The method is clearly presented and, to the best of my knowledge, different to other methods -- although sharing many commonalities (see below). Results improve over a number of small model distillation baselines (although see questions below).

**Weaknesses:**

While to the best of my knowledge the method is different to others -- it is a mix of some existing things (which can be ok). Learning latent vectors by making them predictive of CoT is a pretty standard approach in continuous latent reasoning approaches, e.g. iCoT and COCONUT. In those cases, I believe they show that better latent vectors can be learnt by iteratively removing the CoT to be predicted to compress more of the information, although this is more complex than your approach -- but could it be better?

Other methods also aim to mix latents with CoT text tokens, such as LightThinker (https://arxiv.org/abs/2502.15589) which I believe performs pretty well without needing to keep around a larger and smaller model, I think this work is not mentioned/compared?

Overall, the need to keep both a large and a small model around for inference seems complex, and I think would need to give significant wins to make it worth on it. On the other hand, the counter argument is that frontier models are often these days deployed as mixtures of experts, so this isn't too far off something like that either..

**Questions:**

I don’t know all those baselines well, and they aren’t described in detail (MT-CoT, Step-by-step,  KARD, CasCoD, NesyCD) – although I think MT-CoT would be straight training on the larger models CoT? Did you try on-policy distillation  https://arxiv.org/abs/2306.13649 from the larger model? Or, what about RL of the small model after distillation? Those two variants sounds like some of the strongest standard small model approaches I would try first, and I'm not sure they are compared to?

For which of those baselines do you have to keep around the large model for inference time, I’m assuming many do not, which yours has to, right? So -- this looks like a disadvantage of yours just in terms of complexity...? (That is, the gains would have to be substantial in that case I guess, given the extra complexity).

---

> ### Author Response · Authors · 2025-11-21
> **Response to Reviewer bKFV (part1)**
>
> **We thank the reviewer for the insightful and valuable comments. We respond to each comment as follows and sincerely hope that our rebuttal could properly address your concerns. If so, we would deeply appreciate it if you could raise your score (Rating: 2: reject). If not, please let us know your further concerns, and we will continue actively responding to your comments and improving our submission.**
>
> **Rebuttal: Response to Comments on iCoT / COCONUT / LightThinker and the Two-Model Collaboration Framework**
>
> We sincerely thank the reviewer for the insightful comments. We realize that we may not have sufficiently clarified the fundamental divergence in **design goals** and **mechanisms** in the initial submission. We sincerely hope the following response and new experimental comparisons address your concerns.
>
> Below, we respond in three structured parts:
>
> 1.  **Fundamental Differences:** Why our method is not a variant of iCoT / COCONUT.
> 2.  **Different Objectives:** Why LightThinker (single-model compression) differs from our cross-model transfer.
> 3.  **Justification:** Why two-model collaboration is reasonable and necessary (supported by new scaling experiments).
>
>
> ---
>
> ## **1. Fundamental Differences from iCoT / COCONUT**
>
> We sincerely thank the reviewer for pointing out iCoT and COCONUT. While they also utilize latent states, a systematic comparison reveals that they are **unsuitable substitutes** for our Cross-Model Cognitive Transfer framework due to fundamentally different goals and performance outcomes.
>
> ### **Difference 1: Goals — (Single-model acceleration) vs. (Cross-model ability transfer)**
>
> | Method     | Core Goal                                 | Reasoning Structure            | Requires Explicit CoT? | Requires Large Model at Inference? |
> |------------|--------------------------------------------|--------------------------------|-------------------------|------------------------------------|
> | **iCoT**   | Speed up a *single* model (implicit CoT)   | Forced CoT compression          | ✗                       | No                                 |
> | **COCONUT**| Single-model acceleration (latent-only)    | Implicit compressed reasoning   | ✗                       | No                                 |
> | **Ours**   | Transfer *high-level reasoning ability* from large → small model | Preserve high-level structure; no CoT compression | ✓ (auditable CoT) | Only one short forward pass (latent tokens) |
>
>
> In short:
>
> * **iCoT/COCONUT** aim to reduce the inference cost of a model *by sacrificing the readability and often the accuracy* of the Chain-of-Thought (CoT).
> * **Our Latent Guidance** aims to enable a weak model (e.g., 0.5B–7B) to perform reasoning it *could not originally do*, by borrowing the "thought process" of a stronger model (e.g., 32B–72B).
>
> Therefore, the objectives and design constraints of the methods are entirely different.
>
> ---
>
> ### **Difference 2: Training Paradigm — (Self-compression) vs. (Reasoning-structure transfer)**
>
>
> Our method does **not** try to compress the tokens of a CoT into latent vectors for direct reasoning.
> Instead, latent vectors encode a *high-level plan* that guides the small model.
>
> | Dimension | iCoT / COCONUT                                     | Ours |
> |----------|------------------------------------------------------|-------|
> | Source of reasoning structure | Remove CoT tokens → force all information into latent | Teacher’s high-level cognitive/reasoning plan |
> | Role of latent | Replacement for CoT (implicit CoT) | Latent guidance for the student |
> | Output | No explicit reasoning chain | Student generates full, auditable CoT |
> | Core Function | Self-acceleration | Cross-model ability transfer |
>
> Thus, iCoT/COCONUT are not designed for the setting we address, where **a large model directly enhances a smaller one**.
>
> ---
>
> ### **Difference 3: Performance and Interpretability Differences**
>
> To quantitatively demonstrate this difference, we reproduced iCoT and COCONUT under controlled conditions (using the same Qwen2.5-7B and training data). The results on GSM8K and AQuA (Out-of-Distribution) are revealing:
>
> ---
>
> ### **Table. Supplementary Comparison on Reasoning Performance**
>
> | Method | Model | GSM8K Acc (%) | AQuA (OOD) Acc (%) | CoT Readable? |
> |--------|--------|----------------|----------------------|---------------|
> | Baseline | 7B | 75.4 | 64.7 | ✓ |
> | iCoT | 7B | 54.8 | 39.8 | ✗ |
> | COCONUT | 7B | 57.1 | 40.3 | ✗ |
> | Ours | latent guidance + 7B | **80.5** | **71.7** | ✓ |
>
>
> ---

---

> ### Author Response · Authors · 2025-11-21
> **Response to Reviewer bKFV (part2)**
>
> ### **Key Observations**
>
> We recognize COCONUT and iCoT as pioneering works in latent reasoning. In particular, we applaud **COCONUT for its innovative "continuous thought" paradigm, which enables efficient breadth-first search (BFS) for logical reasoning tasks that require substantial search**. However, our experiments suggest a trade-off between compression and accuracy, observing >10% performance drops on both GSM8K and AQuA when applied to multi-step mathematical reasoning. This highlights a key distinction in design goals: they prioritize single-model acceleration (making an LLM reason faster), whereas our framework focuses on cross-model capability transfer (making an SLM reason better).
>
> We will explicitly discuss this complementary relationship in the final revision, crediting **COCONUT as the foundational work that validated the feasibility of continuous-space reasoning**, thereby conceptually paving the way for our latent guidance approach.
>
> ---
>
> #### **(2) No explicit CoT → No interpretability**
>
> Both iCoT and COCONUT produce **only an answer**, with no reasoning trace.
>
> This makes them unsuitable for domains requiring auditable intermediate reasoning, such as:
>
> - education,
> - medical decision-making,
> - legal reasoning,
> - safety-critical workflows.
>
> In contrast, our method **retains full explicit CoT**: the latent only guides the planning, while the student produces a complete reasoning chain.
>
> ---
>
> **Critical Findings:**
> 1.  **Our Method Improves Performance:** Unlike iCoT which degrades performance, our method **boosts** the 7B model (+5.1% on GSM8K, +7.0% on AQuA) because it injects *additional* cognitive information from the 32B teacher.
> 3.  **Interpretability:** iCoT/COCONUT are "black boxes" (no reasoning trace). Our method produces full, auditable CoT, which is essential for verifying the model's logic.
>
> **Conclusion:** iCoT and COCONUT are optimization techniques for speed, whereas our framework is an enhancement technique for **capability**. They solve different problems.
>
> We hope these supplementary results and expanded discussion help to more clearly position our method in the broader landscape of latent-reasoning approaches.
>
> ---
>
> ## **2. Relation to LightThinker: Single-Model Latent Control vs. Our Two-Model Planning–Execution Decoupling**
>
> LightThinker is an excellent work. We appreciate the reviewer’s pointer to LightThinker. After revisiting the paper and adding controlled experiments, we summarize the key conceptual and empirical distinctions.
>
> ### **What LightThinker Does**
> - **Goal:** Improve *single-model* inference efficiency by teaching the model to “think-and-forget”—retain only essential information, compress the reasoning trace, and reduce runtime.
> - **Mechanism:** Compresses the intermediate reasoning into compact representations (gist tokens) and continues reasoning conditioned on these compressed memories, discarding the original verbose chain.
> - **Objective:** Single-model efficiency optimization (trading minor accuracy drops for memory savings) rather than capability transfer.
>
> ### **What Our Method Does**
> - **High-level cognitive planning is produced by the large model** (Implicit Thinker) and encoded into latent vectors.
> - **The small model acts as an explicit executor**, converting the high-level plan into a full, auditable chain-of-thought.
> - The framework **explicitly decouples planning vs. execution**, enabling *cross-model cognitive transfer* rather than self-compression.
>
> ---
>
> ### **Table. Comparison with LightThinker**
>
> We compared the methods under identical settings as our paper.
>
> | Method | Model | GSM8K | AQuA | Primary Effect |
> |--------|--------|--------|--------|-----|
> | Baseline | 7B | 75.4 | 64.7 |Standard Inference|
> | LightThinker | 7B | 71.1 | 48.8 |Efficiency Gain / Accuracy Trade-off|
> | Ours | latent guidance + 7B | **80.5** | **71.7** |Capability Gain (+5% Avg)|
>
> ### **Key Observations**
> 1. **LightThinker consistently reduces accuracy**, which matches its goal of aggressive reasoning compression.
> 2. LightThinker belongs to the same family as iCoT/COCONUT—**latent compression for single-model acceleration**, rather than cross-model transfer.
> 3. **Our objective is capability transfer, not self-compression**. Therefore the two methods differ in goals, training signals, outputs, and target deployment settings.
>
> At the same time, we noticed that LightThinker maintained excellent performance compared to similar works based on latent reasoning, such as iCOT and COCONUT, with minimal performance degradation. **We will position LightThinker as a strong representative of the “single-model latent reasoning” line of work in the final revision**.

---

> ### Author Response · Authors · 2025-11-21
> **Response to Reviewer bKFV (part3)**
>
> ## **3. On the Complexity of Using Both a Large and a Small Model at Inference**
> We appreciate the reviewer’s concern regarding inference-time complexity. We fully agree that this is not a classical “pure distillation” setup.
> However—as the reviewer rightly pointed out—this is **not far from a mixture-of-experts (MoE) paradigm**, and we believe that our framework can be seen to some extent as *efficient, heterogeneous MoE variant* that delivers quantifiable benefits in practical deployment.
>
> Below we summarize why the two-model design is *practically efficient*, *bandwidth-light*, and *aligned with real deployment scenarios*.
>
> ---
>
>
>
> ## **Value 1: Lower Total Latency Than Either 32B or 7B Baseline**
>
> | Method | Latency (s) | Avg. Reasoning Tokens |
> |---------|--------------|-------------------------|
> | **32B CoT** | 14.4 | 265.8 |
> | **7B baseline** | 7.2 | 291.3 |
> | **Ours (32B→7B)** | **3.5** | **128.9** |
>
> ### **Why our method is faster:**
> - The 32B model does **only one short forward pass** to produce ~5 latent vectors.
> - The 7B model generates **a much shorter reasoning chain** guided by the high-level plan.
> - The dominant runtime cost in LLMs—long autoregressive generation—is *greatly reduced*.
>
> Thus:
>
> - **4.1× faster** than 32B CoT on GSM8K
> - **2.05× faster** than 7B baseline on GSM8K
>
> Even with a large model involved, **total latency is substantially lower**.
>
> ---
>
> ## **Value 2: Ideal for Edge–Cloud Collaboration (On-Device AI)**
>
> Our method naturally fits real deployment workflows where:
>
> - Edge devices (phones/IoT) can host only **0.5B–7B** local models.
> - Cloud servers host **32B–72B** teacher models.
>
> ### **Bandwidth requirement is negligible**
> The teacher sends only **K=5 latent vectors** of dimension 4096 (bf16):
>
> **5 × 4096 × 2 bytes ≈ 40 KB**
>
> Real 4G/5G download rates:
>
> | Network | Realistic Throughput | Latent Transfer Cost |
> |----------|--------------------------|---------------------------|
> | 4G | 20–50 MB/s | 0.8–2 ms |
> | 5G | 100–500 MB/s | ≤0.5 ms (with gzip 2–4× compression) |
>
> Thus, the communication overhead is actually very small, and can even be ignored in a normal network environment.
>
> ### **Advantages**
> - Edge-side sees **no teacher complexity**, but gains *teacher-level reasoning quality*.
> - Perfectly matches industrial needs for hybrid on-device/cloud AI.
>
> ---
>
> ### **Other application scenarios: Flexible API-Level Cost Coordination**
>
> A common deployment pattern is:
>
> - **95%** of queries handled by the 7B model alone.
> - **5%** of difficult queries routed to the 32B model to produce a high-level latent plan, which the 7B executes.
>
> This reduces API cost while maintaining high reasoning quality on tough queries.
>
> ---
>
> ### **Other application scenarios: Latent Caching for High-Frequency Tasks**
>
> For domains like education (math practice, problem banks):
>
> - Latent guidance vectors can be precomputed and cached.
> - Student model runs at **7B-level cost** but achieves **32B-level reasoning planning**.
>
> ---
>
> ## **Value 3: Larger Teacher–Student Gaps Lead to Greater Gains (Scaling Behavior)**
>
> To further evaluate the feasibility of our framework under extreme model-size discrepancies—and to highlight its advantages in such settings—we conducted an additional set of experiments using a **72B teacher → 0.5B/1.5B students** configuration based on Qwen2.
>
> All experimental settings (dataset, preprocessing, training schedule) remained identical to the main paper, with only model sizes changed.
>
> ---
>
> ### **Table. Performance Under Extreme Teacher–Student Gaps**
>
> | Method | GSM8K (%) | Abs ↑ | Rel ↑ | BBH (%) | Abs ↑ | Rel ↑ | Odyssey-Math (%) | Abs ↑ | Rel ↑ |
> |--------|-----------|---------|---------|----------|----------|------------|--------------|-----------|------------|
> | baseline (1.5B) | 39.2 | – | – | 46.9 | – | – | 4.4 | – | – |
> | **Ours (1.5B + latent)** | **50.6** | **11.4** | **29.1%** | **76.2** | **29.3** | **62.5%** | **17.1** | **12.7** | **288%** |
> | baseline (0.5B) | 21.3 | – | – | 23.2 | – | – | 2.8 | – | – |
> | **Ours (0.5B + latent)** | **32.3** | **11.0** | **51.6%** | **71.7** | **48.5** | **209%** | **10.9** | **8.1** | **289%** |
>
> ---
>
> ### **Results and Observations**
>
> #### **1. Consistent and substantial gains across all tasks**
>
> Large improvements are observed in:
> - **Mathematical reasoning** (GSM8K),
> - **Complex logical reasoning** (BBH),
> - **Competition-level mathematical reasoning** (Odyssey-Math).
>
> Notably, the smallest student (0.5B) improves:
> - **Odyssey-Math: 2.8% → 10.9% (+289%)**
> - **BBH: 23.2% → 71.7% (+209%)**
>
> These represent dramatic stability and generalization gains enabled by latent high-level planning.
>
> ---

---

> ### Author Response · Authors · 2025-11-21
> **Response to Reviewer bKFV (part4)**
>
> #### **2. Smaller students benefit even more**
>
> While 1.5B improves by 11.4% on GSM8K:
> - 0.5B improves by 11.0%
> - But on complex tasks (BBH, Odyssey-Math), the **relative gains are substantially larger** for 0.5B.
>
> This strongly matches the expected behavior of our framework:
> > **The smaller the model, the greater the benefit from transferred cognitive planning.**
>
> ---
>
> This may directly supports our claim that the proposed framework acts as an efficient mechanism for **cross-model cognitive transfer**, especially when the student is extremely small relative to the teacher.
>
> We will include these results in the final version and provide a more systematic discussion of scaling laws for large→small transfer.
>
> ---
>
> ## **Ecosystem Summary: Where Our Method Fits**
>
> | Method Class | Representative | Needs LLM at Inference? | CoT Interpretability | OOD Generalization | Suitable for Edge Devices? |
> |----|---|---|---|--|------|
> | Pure Distillation | SFT, NesyCD | No | Yes | Medium–High | No (performance ceiling) |
> | Latent Reasoning | iCoT, Coconut | No | No (black-box) | Low–Medium | No (poor performance/interpretability) |
> | **Ours (Latent Guidance)** | **This work** | **Yes (but very low overhead)** | **Yes** | **High** | **Yes (ideal)** |
>
> ---
>
> ## **Summary**
>
> These additional experiments and analyses help clarify the core positioning of our work:
>
> - It is not a simple variant of distillation, but consistent with the goals of mainstream distillation methods,
> - nor another single-model latent compression method,
> - but a **new, efficient, and scalable large-to-small collaborative reasoning framework**.
>
> This framework achieves a practical *three-way win*:
>
> - **Performance:** surpasses all baselines, especially on OOD tasks.
> - **Efficiency:** total latency is *lower* than the 7B baseline despite using a large model once.
> - **Interpretability:** preserves full, human-auditable reasoning chains.
>
> We appreciate the reviewer’s feedback—it significantly strengthened the clarity and rigor of our work.
>
> ## **4. Reviewer Question: Clarifying the Baselines (MT-CoT, Step-by-step, KARD, CasCoD, NesyCD)**
>
> To avoid any misunderstanding, we provide clear definitions of all baselines included in our comparison.
> All of these methods fall under the **mainstream distillation paradigm**, where a small model performs **standalone inference** after learning from a teacher model.
>
>
> | Baseline | Core Idea | Distillation Paradigm |
> |---|--|-|
> | **Std-CoT** | Directly fine-tune the small model on the teacher’s CoT outputs. | Basic CoT Distillation |
> | **Step-by-step** | Multi-task learning over two targets: rationales (CoT) and final answers. | Multi-task Distillation |
> | **MT-CoT** | Jointly trains on CoT generation and answer prediction. | Multi-task Distillation |
> | **CasCoD** | A cascaded two-stage CoT distillation process. | Cascaded Distillation |
> | **KARD** | Knowledge-augmented distillation that emphasizes generating rationales. | Knowledge-Augmented Distillation |
>
> These baselines share a common goal:
>
> > **Teach a small model to imitate the teacher’s CoT and acquire its reasoning abilities.**
>
> ### **Why compare with them?**
>
> Because they address **the same core problem** as ours:
>
> - ✔ Strengthening small-model reasoning
> - ✔ Using a large model during data construction or training
> - ✔ Attempting to transfer reasoning competence from big to small models
>
>
> Our Latent Guidance framework is introduced precisely to address the limitations of these distillation methods:
>
> - Overfitting to teacher CoT stylistic patterns and **Weak OOD generalization**
> - Unable to transfer Teacher's reasoning ability well, performance is **limited by small model parameters**
>
>
> Therefore, they are **direct and appropriate competitors**.
>
> ---
>
> ## **4.1 Additional Comparison: KD + RL and On-Policy Distillation**
>
>
> Following the reviewer’s suggestion, we conducted supplementary comparisons with standard small-model enhancement techniques, specifically **RL-based refinement** (RSFT, a commonly used Outcome-based RL) and **On-policy distillation** (GKD, following the suggested paper).
>
> #### **Table A: GSM8K Comparison with RL / On-Policy Distillation**
>
> | Method  | Student Model | GSM8K Acc (%) |
> | :- | :-| :-|
> | KD (logits-based)| 7B | 73.0   |
> | KD + RL | 7B  | 78.5   |
> | On-policy distillation | 7B  | 77.6   |
> | **Ours (Latent Guidance)** | latent + 7B   | **80.5** |
>
> **Observations:**
>
> * **Performance:** Our method consistently outperforms both *On-policy distillation* and *RL-based refinement*.
> * **Complementarity:** We view RL as a **powerful and orthogonal** direction. While our method focuses on explicitly transferring high-level reasoning structures from the teacher, RL-based refinement is effective at optimizing the policy through exploration and outcome-based feedback. Thus, the two approaches are **not mutually exclusive**, and integrating Latent Guidance with RL is a promising direction for future work.
>
> ---

---

> ### Author Response · Authors · 2025-11-21
> **Response to Reviewer bKFV (part5)**
>
> ## **4.2 Inference-Time Complexity: Is It Fair? Is It Worth It?**
>
> The reviewer asks a critical question:
> > *Other baselines do not require the large model at inference time. Yours does. Does this introduce extra complexity, and is it justified?*
>
> We provide quantitative evidence to answer this.
>
> ### **Key Clarification: The "Efficiency Paradox"**
>
> At inference, our method requires **only one short forward pass** of the large model (~5 latent tokens). It **does not** require the large model to generate a full CoT.
> Crucially, this high-level guidance leads the student model to produce **much shorter, direct reasoning chains**, dramatically reducing the expensive autoregressive decoding cost.
>
> ---
>
> ### **Latency Experiments (Including the Large-Model Overhead)**
>
> | Dataset | Method | Model Configuration | Accuracy (%) ↑ | Tokens | Latency (s) ↓ |
> |---------|--------|----------------------|----------------|--------|---------------|
> | **GSM8K** | Large Model | Qwen2.5-32B-Instruct | 84.1 | 265.8 | 14.4 |
> | | **Ours** | 32B + 7B | 80.5 | **128.9** | **3.5** |
> | | Small Model | Qwen2.5-7B-Instruct | 75.4 | 291.3 | 7.2 |
> | **ARC-C** | Large Model | Qwen2.5-32B-Instruct | 95.5 | 280.7 | 15.2 |
> | | **Ours** | 32B + 7B | 90.9 | **138.3** | **3.6** |
> | | Small Model | Qwen2.5-7B-Instruct | 77.0 | 293.8 | 7.2 |
> | **SVAMP** | Large Model | Qwen2.5-32B-Instruct | 86.2 | 178.8 | 9.8 |
> | | **Ours** | 32B + 7B | 85.9 | **68.6** | **2.2** |
> | | Small Model | Qwen2.5-7B-Instruct | 79.1 | 216.2 | 5.3 |
>
> ### **Interpretation: Complexity is Justified by Speed and Quality**
> 1.  **Faster than the Small Model:** By cutting the generated token count by >50% (e.g., 291 $\to$ 128 on GSM8K), the reduction in decoding time outweighs the cost of the single large-model forward pass.
> 2.  **Net Result:** Total latency is **~2x lower** than the 7B baseline, despite the "extra complexity" of the teacher.
>
> This demonstrates that the need for a large model is **not an overhead**; the overall system remains efficient and even faster than a single small model.
>
> ---
>
>
> ### **Final Clarification for the Reviewer’s Question**
>
> We fully agree with the reviewer that many baselines do not require the large model at inference time.
> However, in our design:
>
> - The large model contributes only **one short forward pass**(5 latent tokens), not full reasoning.
> - The resulting high-level plan enables the student to perform **shorter, more focused CoT generation**.
> - Consequently, **total latency remains below the 7B baseline**.
>
> **We humbly hope our response has addressed your concerns. If you have any additional concerns or comments that we may have missed in our responses, we would be most grateful for any further feedback from you to help us further enhance our work.**

---

> ### Author Response · Authors · 2025-11-27
>
> Dear Reviewer bKFV,
>
> We are deeply grateful for the time and effort you have invested in reviewing our paper. Your valuable questions and insights have significantly contributed to enhancing our work.
>
> To address your queries and concerns, we have meticulously prepared detailed responses. We fully recognize the multitude of submissions you juggle, and we genuinely value the attention you give to our research. Acknowledging the hectic schedule during the discussion period, **we summarize here the key points of our rebuttal to facilitate a quick read and conserve your valuable time**.
>
> * **Fundamental Difference from iCoT/COCONUT/LightThinker:** We provided comparative experiments showing that iCoT/COCONUT are primarily designed for acceleration and can suffer from performance degradation (observing >10% accuracy drops on both GSM8K and AQuA). In contrast, our method focuses on **cross-model capability transfer**, consistently **boosting** student performance (e.g., +5.1% on GSM8K and +7.0% on AQuA). Furthermore, unlike latent-only approaches, our method **preserves the full Chain-of-Thought (CoT)**, ensuring the reasoning process remains readable and auditable (Please refer to **Rebuttal Part 1 & 2**).
> * **Clarification on Inference Latency & Complexity:** We provided latency breakdowns demonstrating that despite using a teacher model, our **total end-to-end latency is actually ~2x lower** than the standalone 7B baseline. This is because the high-level guidance enables the student to generate significantly more concise reasoning chains, offsetting the minimal teacher overhead (Please refer to **Rebuttal Part 3**).
> * **Broad Application Scenarios:** We expanded the discussion on how this large-small collaborative framework fits into practical deployment scenarios, such as **Edge-Cloud collaboration** and efficient dynamic routing, further justifying the practical value of the dual-model setup (Please refer to **Rebuttal Part 3**).
> * **New Experiments on Extreme Scaling (72B Teacher → 0.5B Student):** We added experiments showing massive gains when a tiny 0.5B model is guided by a 72B teacher. For instance, on the challenging **BBH benchmark**, the 0.5B student achieved an **absolute gain of +48.5%** (relative improvement of **+209%**), validating the necessity and effectiveness of the framework in extreme compression settings (Please refer to **Rebuttal Part 3**).
> * **Comparison with RL & On-Policy Distillation:** As requested, we compared our method with standard KD+RL and On-Policy Distillation baselines, showing that Latent Guidance consistently yields superior accuracy (Please refer to **Rebuttal Part 4**).
>
> We humbly hope that our responses and new experimental evidence have adequately addressed your concerns. We are eager to address any additional queries you might have.
>
> We are fully aware of your busy schedule, especially during this discussion period. We understand that reviewing numerous rebuttals can be extremely challenging. As the deadline for the author-reviewer discussion period is approaching, we are looking forward to your feedback.
>
> We would deeply appreciate it if you could reconsider your evaluation and raise your score if our rebuttal has addressed your concerns. If not, please let us know your further concerns, and we will continue actively responding to your comments.
>
> Thank you for your invaluable time and consideration.
>
> Best regards,
>
> Authors

---

### Official Review · Reviewer_bGgN · 2025-10-30

**Soundness:** 3
**Presentation:** 2
**Contribution:** 3
**Rating:** 6
**Confidence:** 3

**Summary:**

This paper addresses the high computational cost of complex reasoning in LLMs. The authors argue this inefficiency stems from the tight coupling of "high-level cognitive planning" (strategy) and "low-level linguistic realization" (text generation). To solve this, they propose "Latent Guidance," a collaborative framework that decouples these processes. The framework uses two models: an **Implicit Thinker** (a large LLM, e.g., Qwen2.5-32B) processes the input question and special "thought tokens" to produce a set of compact "latent guidance vectors" ($H_{guidance}$); an **Explicit Executor** (a small LLM, e.g., Qwen2.5-7B) receives both the question and $H_{guidance}$ to autoregressively generate the final reasoning chain and answer. The core of this "Cognitive Distillation" is a dual-loss training objective for the large model. It includes a task loss for the correct answer, and, crucially, a reconstruction loss $\mathcal{L}\_{recon}$ that compels the latent vectors $H_{guidance}$ to contain enough information to reconstruct the entire original, detailed reasoning chain $R$. The authors provide a theoretical analysis showing that this $\mathcal{L}\_{recon}$ maximizes the mutual information between the plan and the reasoning chain. Experiments across eight benchmarks show the 7B model, when guided, gains up to 13.9% in accuracy, runs 2x faster than its 7B baseline, and is 4x faster than the 32B teacher, while also showing superior OOD generalization.

**Strengths:**

1.  The paper's primary strength is its novel "Cognitive Distillation" framework. The conceptual move to decouple planning from realization is intellectually appealing. It reframes distillation from mimicking *outcomes* (final text) to transferring a *process* (a high-level plan), which is a contribution to the field.

2.  The $\mathcal{L}\_{recon}$ loss is not merely empirical; it is rigorously justified with an information-theoretic analysis in Appendix A. The derivation showing that minimizing $\mathcal{L}\_{recon}$ is equivalent to maximizing the mutual information $I(R;H_{guidance}|Q)$ provides a solid theoretical foundation for why the latent vectors should be information-rich.

3. The method achieves strong results, particularly in OOD generalization (Table 1). The fact that it consistently outperforms strong distillation baselines (like NesyCD) on unseen datasets lends weight to the hypothesis that distilling a "strategy" is more robust and generalizable than distilling text.

4.  The results in Table 6 are noteworthy. Achieving a 2x speedup over a 7B baseline while simultaneously boosting its accuracy by +13.9% (on ARC-C) represents a clear Pareto improvement and a practical achievement.

**Weaknesses:**

1. The paper's central thesis—that $H_{guidance}$ represents an *abstract, high-level "cognitive plan"* —is not sufficiently supported by the provided evidence.
    *   **t-SNE Visualization (Fig. 3):** This evidence is weak. t-SNE is known for creating illusory clusters. The observed clusters, which are given post-hoc labels like "Sequential Multi-Step Reasoning," are far more likely to be simple high-dimensional clusters of "problem types" (e.g., all two-step addition problems) rather than proof of abstract, disembodied "strategies".
    *   **Probing Analysis (Table 5):** This analysis is tautological. The $\mathcal{L}\_{recon}$ loss *explicitly trains* $H_{guidance}$ to be able to reconstruct the full text. Therefore, "proving" that a probe can predict properties of that text (like "Reasoning Step Count" or "Mathematical Operator")  from $H_{guidance}$ is not evidence of a cognitive plan; it is merely a sanity check that the loss function *worked*.

2. The ablation in Table 4 shows that using $K=10$ thought tokens yields better performance on GSM8K (81.1%) than the $K=5$ (80.5%) used for the main results. This suggests the paper may not be reporting its strongest possible configuration.

3. The qualitative analysis in Table 3 relies entirely on GPT-4o as an evaluator (Appendix E). This is not a sufficient substitute for human evaluation (or stronger models like GPT-5), especially when judging nuanced qualities like "Relevance."

**Questions:**

See weaknesses

---

> ### Author Response · Authors · 2025-11-21
> **Response to Reviewer bGgN (part1)**
>
> **We thank the reviewer for the insightful and valuable comments. We respond to each comment as follows and sincerely hope that our rebuttal could properly address your concerns. If so, we would deeply appreciate it if you could raise your score. If not, please let us know your further concerns, and we will continue actively responding to your comments and improving our submission.**
>
>
> ---
>
> ## **1. Is $H_{guidance}$ truly a "Cognitive Plan"? (Response to Weakness 1)**
>
> We sincerely thank the reviewer for this important critique. We fully agree that the original t-SNE visualization and probing results are not sufficient on their own to conclude that $H_{guidance}$ represents an abstract “cognitive plan” rather than a compressed textual embedding.
>
> Your comments prompted us to conduct additional analyses explicitly designed to verify this hypothesis: **Cross-Lingual Control Analysis:** Keeping the *reasoning strategy fixed* while making *large changes to surface text*.
>
>
> ---
>
> ## Cross-Lingual Control Analysis: Strategy Fixed, Text Perturbed
>
> **Hypothesis**
>
> * If $H$ is primarily a **textual compression**, switching languages (e.g., English $\to$ Chinese) should drastically change the embedding due to the complete shift in vocabulary and token space.
> * If $H$ is a **strategy representation**, as long as the reasoning *structure* is unchanged, the embedding should remain similar regardless of language.
>
> ---
>
> **Experimental Design**
>
> We explicitly filtered and selected **50 math word problems** (e.g., age problems, rate problems, chicken-and-rabbit problems) that are amenable to **multiple distinct solution methods**. For each problem, we enforced the **Teacher (Implicit Thinker)** to generate specific types of reasoning chains via strict prompting constraints:
>
> **Group A: Fixed Strategy (Algebraic), Varying Surface Form**
>
> 1.  **English ($R_{En}$):** Standard English CoT using algebraic equations (variables $x, y$).
> 2.  **Chinese ($R_{Zh}$):** The exact same algebraic logic, but realized in Chinese text. (Note: This represents a maximal shift in token space).
> 3.  **Paraphrase ($R_{En-Para}$):** English CoT with identical logic but distinct phrasing.
>
> **Group B: Fixed Language (English), Varying Strategy**
>
> 4.  **Different Strategy ($R_{En-Enum}$):** Solved using an "Arithmetic/Enumeration" method (step-by-step calculation without variables) instead of Algebra.
>
> We then measured the **Cosine Distance** and **L2 Distance** between the resulting latent guidance vectors (sequence-averaged) generated by the Teacher.
>
> **Table. Latent Guidance Vector Distances (Cosine / L2)**
>
> | Comparison Pair | Condition | Avg. Cosine Distance ↓ | Avg. L2 Distance ↓ | Interpretation |
> | :--- | :--- | :---: | :---: | :--- |
> | **En vs En-Para** | Same Strategy, Diff Phrasing | 0.016 | 75.67 | Paraphrasing has almost no effect. |
> | **En vs Zh** | Same Strategy, Diff Language | 0.095 | 177.79 | Language change has moderate effect. |
> | **En vs En-Enum** | Diff Strategy, Same Language | **0.159** | **234.21** | **Strategy change causes largest shift.** |
>
> *(Comparison: The shift caused by changing strategy (0.159) is ~1.7x larger than the shift caused by completely changing the language (0.095).)*
>
>
> ---
>
> **Interpretation**
>
> These results suggest that:
>
> - $H_{\text{guidance}}$ is **highly sensitive to strategy changes**,
> - but **remarkably robust to language/surface-form differences**.
>
> From a text-compression perspective:
>
> - Switching English → Chinese should drastically change the embedding.
>
> From a strategy-representation perspective:
>
> - As long as the reasoning *structure* is unchanged, the embedding should remain similar.
>
> The observed behavior aligns more naturally with the latter.
> This provides initial evidence that $H$ is **not primarily a text-style embedding**,
> but rather a **strategy-centered representation**.
>
>
> **Summary**
>
> Admittedly, interpreting deep latent representations remains a challenging open problem. We cannot claim with certainty that the latent vectors are purely abstract "plans" without any textual residue. However, these experimental results—showing that the representation is more sensitive to strategy changes than to language switches—provide consistent signals that our framework encourages the model to encode higher-level organizational information rather than just surface text. We hope this analysis helps alleviate the reviewer's concern regarding the nature of the latent guidance.
>
> ---

---

> ### Author Response · Authors · 2025-11-21
> **Response to Reviewer bGgN (part2)**
>
> ## **2. Ablation: Why K=5 instead of K=10? (Response to Weakness 2)**
>
> We sincerely thank the reviewer for this insightful observation. You correctly noted that using more thought tokens (K=10) yields slightly higher GSM8K accuracy (81.1%) compared to K=5 (80.5%).
>
> Our goal is **not** to push for the absolute highest accuracy at any cost, but to achieve a **balanced and deployable trade-off**.
>
> ---
>
> ### **Reason 1. K=10 provides small accuracy gains but nearly doubles the computational cost on teacher model**
>
> Increasing the number of latent vectors from 5 to 10 results in:
>
> - ~2× larger latent payload,
> - ~2× more computation in the projection layer,
> - ~2× more computation in the teacher’s forward pass.
>
> Thus:
>
> **K=10 improves GSM8K by only +0.6%, but increases inference cost by roughly 2× on teacher model.**
>
> ---
>
> ### **Reason 2. Mutual information (MI) analysis shows diminishing returns beyond K=5**
>
> ### Mutual Information vs Number of Latent Thoughts (K)
>
> | K (Latent Tokens) | MI (nats) | Marginal Gain |
> | :--- | :---: | :---: |
> | **3** | 2.71 | - |
> | **5** | 2.87 | **+0.16** (Significant) |
> | **10** | 2.94 | **+0.07** (Diminishing) |
>
> **Observation:**
> - The MI gain **drops by more than half** when increasing K from 5 to 10.
> - **K = 5 captures ~95% of the MI achieved by K = 10**, while requiring only half the computational cost.
>
> This matches the observation that accuracy only increases marginally from 80.5% (K=5) to 81.1% (K=10).
> Beyond K=5, the latent carries only slightly more useful information, yet costs much more to compute.
>
> ---
>
> **K=5 is the “sweet spot” for performance–efficiency balance**
>
>
> K=10 remains a valid configuration for scenarios where **efficiency is not a concern** and users want the *highest accuracy possible*, but it does not align with the main goal of our framework:
> **efficient large→small cognitive planning transfer.**
>
> ---
>
> **We will clearly state this design rationale**
>
> - K=5 is chosen as the main configuration due to its strong performance–efficiency trade-off,
> - K=10 can achieve slightly higher scores, but with disproportionately higher cost,
> - our method aims to enable efficient cross-model reasoning rather than maximize accuracy at any cost.
>
> We thank the reviewer again for highlighting this point; it helps us make our design choices clearer and more transparent.

---

> ### Author Response · Authors · 2025-11-21
> **Response to Reviewer bGgN (part3)**
>
> ## **3. Qualitative Evaluation: GPT-4o vs Stronger Judges (Response to Weakness 3)**
>
>
> We agree that relying solely on GPT-4o might be insufficient. To address this, we re-ran the ELI5-Test evaluation using **GPT-5**, a significantly stronger reasoning model, employing the same reference-anchored protocol.
>
>
> ---
>
> ## **Table R4. ELI5-Test Qualitative Evaluation (Evaluator = GPT-5)**
>
> | Category | SFT Corr. | SFT Rel. | SFT Avg | KD Corr. | KD Rel. | KD Avg | Ours Corr. | Ours Rel. | Ours Avg |
> | :--- | :--- | :--- | :--- | :--- | :--- | :--- | :--- | :--- | :--- |
> | Biology | 7.55 | 7.75 | 7.65 | 7.42 | 7.74 | 7.58 | 7.92 | 8.38 | 8.15 |
> | Chemistry | 7.48 | 7.72 | 7.60 | 7.33 | 7.78 | 7.55 | 7.95 | 8.45 | 8.20 |
> | Culture | 7.92 | 8.38 | 8.15 | 7.61 | 7.84 | 7.73 | 8.16 | 8.34 | 8.25 |
> | Earth Science | 7.39 | 8.01 | 7.70 | 7.05 | 7.41 | 7.23 | 8.24 | 8.56 | 8.40 |
> | Economics | 7.71 | 8.00 | 7.85 | 7.67 | 7.93 | 7.80 | 8.32 | 8.75 | 8.53 |
> | Engineering | 7.94 | 8.41 | 8.18 | 7.69 | 7.87 | 7.78 | 8.07 | 8.49 | 8.28 |
> | Mathematics | 7.80 | 8.10 | 7.95 | 7.98 | 8.27 | 8.13 | 8.59 | 8.81 | 8.70 |
> | Other | 7.36 | 7.83 | 7.60 | 7.29 | 7.47 | 7.38 | 7.90 | 8.39 | 8.15 |
> | Physics | 7.69 | 8.07 | 7.88 | 7.47 | 7.63 | 7.55 | 8.18 | 8.63 | 8.40 |
> | Psychology | 7.46 | 7.71 | 7.58 | 8.02 | 8.39 | 8.20 | 8.28 | 8.97 | 8.63 |
> | Repost | 7.60 | 8.08 | 7.85 | 7.32 | 7.48 | 7.40 | 8.04 | 8.56 | 8.30 |
> | Technology | 7.74 | 7.91 | 7.83 | 7.89 | 8.21 | 8.05 | 8.28 | 8.94 | 8.61 |
> | **Overall Avg** | **7.64** | **8.00** | **7.82** | **7.56** | **7.84** | **7.70** | **8.16** | **8.61** | **8.38** |
>
> ---
>
> ## **Interpretation of New GPT-5 Results**
>
> - GPT-5 is **more conservative**, giving lower absolute scores than GPT-4o.
> - However, the **relative trend is unchanged**—and in many categories becomes **even clearer**:
>   - Our method achieves consistently higher grounded scores across **all 12 topical domains**.
>   - Improvements occur both in **correctness** and **relevance**, suggesting stronger factual grounding and better alignment with the gold answer.
>
> These findings reinforce our earlier conclusions about the qualitative robustness and OOD generalization of our method. This confirms that our qualitative advantage is robust and not an artifact of a specific evaluator.
>
> Once again, we thank the reviewer for this valuable suggestion—it significantly improved the rigor of our evaluation.
>
>
>
> **We humbly hope our response has addressed your concerns. If you have any additional concerns or comments that we may have missed in our responses, we would be most grateful for any further feedback from you to help us further enhance our work.**

---

> > ### Comment · Reviewer_bGgN · 2025-11-27
> > **Thanks to authors' responses**
> >
> > Thanks to the authors' detailed responses, especially the experiment and analysis about $H_{guidance}$, these responses address most of my concerns, thus I decided to increase my score by 2 points.

---

> > > ### Author Response · Authors · 2025-11-27
> > >
> > > Dear Reviewer bGgN,
> > >
> > > Thank you very much for your encouraging feedback. We are deeply grateful for your confirmation that we have successfully addressed your concerns.
> > >
> > > Your thoughtful evaluation and constructive insights regarding the latent plan analysis were invaluable to improving our work. We sincerely appreciate your decision to raise your score based on our new experiments and clarifications.
> > >
> > > Thank you once again for your time and support.
> > >
> > > Best regards,
> > >
> > > Authors

---

### Official Review · Reviewer_gdf4 · 2025-11-01

**Soundness:** 3
**Presentation:** 4
**Contribution:** 3
**Rating:** 6
**Confidence:** 3

**Summary:**

The paper proposes a novel framework named "Latent Guidance" (LG) designed to empower small language models (SLMs) with the complex reasoning abilities of large language models (LLMs). The authors term this new paradigm "Cognitive Distillation," which decouples high-level cognitive planning from low-level linguistic realization. In this framework, a large "Implicit Thinker" model performs the complex planning and compresses its strategy into a set of latent vectors ($H_{guidance}$). A small "Explicit Executor" model then receives this guidance and generates the final textual reasoning chain. The framework is optimized via a dual-loss training objective.

The authors prove their method's effectiveness and efficiency via both empirical experiments and theoretical analysis. Comprehensive ablations like the probing experiments of the in-depth analysis of the latent cognitive plan indicates the small model executes a structured, abstract plan, rather than just learning feature correlations.

**Strengths:**

1. The manuscript is well-organized and easy to follow.

2. The proposed idea of cognitive distillation is sound.  The information theoretical analysis is solid.

3. The appendix includes almost all theoretical and experimental details like the training hyperparameters and computational resources.

4. The paper goes beyond accuracy tables to explain the inner mechanisms. The t-SNE visualizations and quantitative probing tasks provide compelling evidence that the $H_{guidance}$ vectors are not just feature containers but successfully encode high-level, abstract reasoning strategies (e.g., "Sequential Multi-Step Reasoning").

**Weaknesses:**

1. There is a contradiction between the core methodological design and the empirical results.
 The reconstruction loss  and  SLM loss explicitly train the models to reconstruct the original, full-length reasoning chain.
However, the paper claims a key benefit is that the SLM can generate substantially more concise reasoning chains (e.g., 128.9 tokens vs. SFT's 235.4). The paper cannot train the model to do one thing (reconstruct the full 235-token chain) and then claim its success for doing the opposite (generating a 129-token chain) without a clear explanation. Is conciseness an emergent property? Or is the method description in Eq 6 inaccurate?

2. The paper calls L_recon the "cornerstone" of the method. However, the ablation study (Table 4) shows that removing it causes only a minor performance drop (e.g., -2.3% on GSM8K). This small drop does not support its "cornerstone" status for in-domain performance.


3. Line 152, Line 157: the author should use correct right single quote.

**Questions:**

1. About the latency in Table 6, does the reported time for "Ours" represent the total end-to-end latency ($T_{total} = T_{step\_{(32B\_{fwd})}} + T_{step\_{(7B\_{gen})}}$)?

2. See weakness 1

3. Have the authors considered an additional baseline that involves providing the full text of the 32B teacher's reasoning chain ($R_{LLM}$) directly into the 7B small model's prompt at inference time?

---

> ### Author Response · Authors · 2025-11-21
> **Response to Reviewer gdf4 (part1)**
>
> **We thank the reviewer for the insightful and valuable comments. We respond to each comment as follows and sincerely hope that our rebuttal could properly address your concerns. If so, we would deeply appreciate it if you could raise your score. If not, please let us know your further concerns, and we will continue actively responding to your comments and improving our submission.**
>
>
>
>
> ## **1. Mechanism Analysis: Shorter Reasoning Chains (Response to Weakness 1)**
>
> We sincerely thank the reviewer for raising this important question regarding why the student model produces **shorter reasoning chains** than other baselines. We fully understand the concern and agree that this phenomenon warrants careful examination.
>
>
> To better understand and explain this behavior, we conducted several additional analyses during the rebuttal period, including:
>
> - **Redundancy Ratio Analysis**
> - **Gradient Flow Contribution Analysis**
> - **Logic Density Analysis**
> - **Perplexity (PPL) Preference Analysis**
>
>
> Below, we walk through each analysis in detail.
>
>
> ---
>
> ## 1.1 Redundancy Ratio Analysis: How teacher and student CoT differ
>
> To obtain a clearer picture of stylistic differences, we sampled 500 CoT instances from the test set and conducted a coarse-grained token-level categorization. Using a combination of large-model heuristics and rule-based detection, we categorized tokens into three types:
>
> - **Core**: directly related to reasoning steps (numbers, operators, variables, formulaic structure, key logical connectives such as *if/then/therefore*).
> - **Exploratory**: “thinking aloud” expressions such as *let’s try…*, *maybe…*, *let’s check again*, which reflect the teacher’s exploratory problem-solving behavior.
> - **Redundant**: reader-oriented natural-language decoration such as *note that…*, *we can see that…*, *in other words…*.
>
>
> The average distributions are:
>
> ### Teacher (Qwen2.5-32B)
> | Token Type | Avg. Ratio | Examples |
> |------------|------------|----------|
> | **Core** | **57.6%** | “solve”, “compute”, “x”, “13^2” |
> | **Exploratory** | 18.1% | “let’s try…”, “maybe…”, “check again” |
> | **Redundant** | 24.3% | “note that…”, “we can see that…” |
>
> ### Student (7B) under Latent Guidance
> | Token Type | Avg. Ratio |
> |------------|------------|
> | **Core** | **85.6%** |
> | **Exploratory** | **3.9%** |
> | **Redundant** | **10.5%** |
>
> These observations highlight a clear pattern:
>
> - The student’s responses collapse toward **pure reasoning steps**.
> - The teacher’s CoT contains a large amount of *exploratory* and *reader-facing* language.
>
> We believe this difference arises because latent guidance provides **a complete high-level reasoning blueprint**.
> Thus, the student **no longer needs to explore**, and instead simply executes the pre-structured reasoning path with minimal linguistic overhead.
>
>
> ---
>
> ## 1.2 Gradient-Based Token Contribution Analysis
>
>
> To understand what the model actually learns, we tracked the gradient norms associated with the reconstruction branch. We computed gradient norms for token embeddings across the same three categories.
>
> **Avg. Gradient Norm During Training:**
>
> | Token Type | Avg. Gradient Norm | Interpretation |
> |------------|---------------------|----------------|
> | **Core** | **0.0077** | Gradients concentrate on the true reasoning structure |
> | – variable | 0.0082 | Variable dependency structure |
> | – operator | 0.0057 | Operation structure forming the reasoning backbone |
> | – number | 0.0029 | Numeric information as auxiliary |
> | **Exploratory** | **0.0022** | Some contribution but clearly lower |
> | **Redundant** | **0.0009** | Very limited gradient impact |
>
> These results suggest the following:
>
> - Although redundant tokens appear frequently in the teacher’s chain,
>   **their gradient contributions are dramatically smaller**.
> - Exploratory tokens have moderate but noticeably lower influence.
> - **Core reasoning tokens dominate parameter updates**, shaping what the student learns most strongly.
>
> In other words:
>
> > During training, the student *selectively absorbs* the teacher’s information:
> > it strongly internalizes **reasoning structure**, while largely *ignoring* stylistic redundancy.
>
> This pattern may indicate that the student tends to focus more on learning the core reasoning structure rather than the teacher’s exploratory or stylistic tokens. Such selective emphasis may help explain why the student later produces shorter reasoning paths while maintaining correctness.
>
>
> ---

---

> ### Author Response · Authors · 2025-11-21
> **Response to Reviewer gdf4 (part2)**
>
> ---
>
> ## 1.3 Logic Density Analysis: Shorter Chains but Comparable Reasoning Depth
>
> We analyzed the number of reasoning steps and the number of generated tokens across three paths:
>
> - **(R_teacher):** the 32B teacher’s original CoT on test dateset
> - **(R_baseline):** the 7B SFT baseline’s original CoT on test dateset
> - **(R_concise):** the concise reasoning chains produced by our 7B student model under latent guidance on test dateset
>
> ### Table. Average Length and Logic Density
> | Path | Avg. Tokens | Avg. Reasoning Steps | Logic Density = steps/token |
> |------|-------------|----------------------|------------------------------|
> | **(R_teacher)** | 265.8 | 5.6 | 0.022 |
> | **(R_baseline)** | 291.3 | 4.7 | 0.016 |
> | **(R_concise)** (Ours) | **128.9** | **5.5** | **0.042 (highest)** |
>
> We observe that:
>
> - The **number of reasoning steps** in *(R_concise)* is nearly identical to that of the teacher.
> - The **token count** is significantly smaller.
> - As a result, **logic density** roughly doubles.
>
> This pattern may suggest that the student has learned to follow a reasoning structure similar to the teacher’s—given that their step counts are closely aligned. Together with Analyses 1 and 2, this could indicate that the student selectively omits redundant tokens during inference, resulting in a more concise expression of the same underlying reasoning process.
>
>
>
> ---
>
> ## 1.4 Perplexity (PPL) Analysis: The Student Has Not Forgotten the Teacher’s Long Chains
>
> To test whether the student has “forgotten” the teacher’s longer chains, we evaluated perplexity on three types of reasoning paths.
>
> - **(R_teacher):** the 32B teacher’s original CoT on test dateset
> - **(R_baseline):** the 7B SFT baseline’s original CoT on test dateset
> - **(R_concise):** the concise reasoning chains produced by our 7B student model under latent guidance on test dateset
>
> ### Table. PPL Comparison (lower is better)
> | Path | Description | Avg. PPL ↓ | Interpretation |
> |------|-------------|-------|----------------|
> | **(R_concise)** (Ours) | Student’s concise chain | **2.3** | Most preferred by the student |
> | **(R_teacher)** | Teacher’s original long chain | **2.7** | Still low; student has *not* forgotten it |
> | **(R_baseline)** | Baseline’s long exploratory chain | **6.2** | Much less natural to the student |
>
> Key observations:
>
> ### 1. The student has *not* forgotten the teacher’s long chain.
> - *(R_teacher)* still has **very low PPL**, indicating the student still judges it as natural and well-formed.
> - This means that if forced, students *can* generate/understand it.
>
> ### 2. The student *prefers* concise chains.
> - *(R_concise)* achieves the lowest PPL.
>
> ### 3. The baseline’s long exploratory chains are strongly disfavored.
> - The high PPL for *(R_baseline)* suggests that long reasoning chain is unnatural to the student.
>
> This PPL analysis is consistent with the redundancy, gradient, and logic-density findings.
>
> ---
>
> ## Summary and Response to the Reviewer’s Concern
>
> Across these analyses, several patterns emerge that may help explain the reviewer’s observation:
>
> - **Training objective:**
>   The framework appears to encourage the transfer of the teacher’s underlying reasoning structure rather than the surface-level linguistic form.
>
> - **During training:**
>   Gradient contributions tend to focus more on core reasoning elements, while redundant or exploratory tokens show noticeably lower influence on parameter updates.
>
> - **During inference:**
>   When guided by high-level latent representations, the student often generates reasoning chains that retain similar step structures but use more compact language.
>
> While these findings do not definitively explain every aspect of this behavior, they offer signals suggesting why the student might favor shorter reasoning chains under our framework. We acknowledge that further investigation—such as finer-grained annotations or larger-scale studies—would be valuable for deepening this understanding.
>
> Our intention with these analyses is to help alleviate the reviewer’s concerns by providing possible explanations grounded in empirical observations. We sincerely appreciate the reviewer’s feedback and will continue refining both the experiments and the discussion.

---

> ### Author Response · Authors · 2025-11-21
> **Response to Reviewer gdf4 (part3)**
>
> ## **2. Is $\mathcal{L}_{recon}$ really a "Cornerstone"? (Response to Weakness 2)**
>
>
> We sincerely thank the reviewer for the sharp and constructive observation.
> Your comment highlights an important point regarding the role of the reconstruction loss.
> Below, we respond in detail.
>
>
> ---
>
> ## 2.1 Why -2.3% is Significant
>
> In our ablation study (Table 4), the removal of $\mathcal{L}_{recon}$ results in the **single largest performance drop** among all architectural ablations. While 2.3% may seem small on in-domain tasks (where the student might guess the answer via shortcuts), the impact is more profound on the **quality and density** of the transferred information.
>
>
> Without $ \mathcal{L}_{\text{recon}} $:
>
> - the task loss $ \mathcal{L}_{\text{task}} $ alone already forces the teacher to put some “answer-relevant” information into the latent guidance
> - but $ \mathcal{L}_{\text{task}} $ optimizes only *the final answer* (the **what**), not the reasoning process (the **how**)
>
> The reconstruction loss is what makes
> **$H_{\text{guidance}}$ a high-fidelity carrier of the full reasoning structure**, not just the minimal signal required for correctness.
>
>
> ---
>
> ## 2.2 Information Density Analysis
>
> To prove this, we measured the Mutual Information (MI) between the latent guidance and the ground-truth reasoning chain ($R$), with and without $\mathcal{L}_{recon}$.
>
> **Table: Impact of Reconstruction Loss on Latent Information**
>
> | Training Setup | Estimated MI (nats) | Relative Gain | Interpretation |
> | :--- | :---: | :---: | :--- |
> | **w/o $\mathcal{L}_{recon}$** | 1.72 | — | Serious information loss, possibly encodes mostly "Answer" cues. |
> | **w/ $\mathcal{L}_{recon}$ (Ours)** | **3.10** | **+80.2%** | Information is clearly rich |
>
> **Observation:**
> * Without $\mathcal{L}_{recon}$, the MI ($\approx 1.72$) suggests the latent may only capture minimal cues necessary for the answer (Task Loss).
> * With $\mathcal{L}_{recon}$, the MI jumps to **3.10** (matching the entropy of the reasoning chain).
> * **Conclusion:** $\mathcal{L}_{recon}$ is the specific component that forces the latent to carry the **"How" (Process)** rather than just the **"What" (Result)**. This structural fidelity is the "cornerstone" of our method's OOD generalization capability.
>
> ---
>
> ## **3. Clarifications (Typos and Latency)**
>
> **Typos:** Thank you for pointing out the quote mark issues (Line 152, 157). We will correct them in the final revision.
>
> ---
>
> ## **4. About Table 6 Latency: Does “Ours” report full end-to-end time?**
>
> **Yes — absolutely.**
>
> The latency reported for “Ours” *includes both*:
>
> 1. **5 latent tokens** of the 32B Implicit Thinker
>    (≈ 540 ms/token-equivalent)
>
> 2. **Full COT** of the 7B Explicit Executor
>    (≈ 240 ms/token)
>
> We will explicitly clarify this definition in the caption of Table 6.
>
> ---
>
> ## **5. Additional Baseline: Feeding the full 32B CoT into the 7B model**
>
> This is an excellent suggestion to benchmark the "upper bound" of transfer. We added a baseline **"Text-Prompt (32B CoT $\to$ 7B)"**, where the 7B model is prompted with the full reasoning chain generated by the 32B teacher.
>
>
> ### **Text-Prompt (32B CoT → 7B)**
>
> | Method | GSM8K | BBH | AGIEval | ARC-E | ARC-C | Odyssey | AQuA |
> |---|---|----|------|------|-------|------|-----|
> | Text-Prompt (32B CoT → 7B) | 84.1 | 85.1 | 60.0 | 96.5 | 95.1 | 22.1 | 73.4 |
> | SFT | 77.3 | 79.2 | 47.1 | 94.5 | 89.4 | 15.5 | 65.4 |
> | KD | 73.0 | 77.6 | 55.8 | 89.1 | 83.2 | 17.9 | 69.5 |
> | **Ours** | **80.5** | **79.8** | **56.7** | **96.4** | **90.9** | **22.7** | **71.7** |
>
> ### **Interpretation**
>
> While “Text-Prompt (32B CoT → 7B)” performs strongly, it is *not* a practical baseline:
>
> - The 32B model must **first generate the full CoT autoregressively**,
>   which costs **14.4 seconds** and **~266 tokens** on GSM8K alone.
> - Then the 7B model must process this long prompt and generate an answer.
> - Total cost:
>   **(32B CoT generation) + (7B encoding) + (7B generation)**
>
> This is **more expensive than simply using the 32B model directly**,
> and the 7B model contributes nothing beyond extra latency.
>
> In contrast:
>
> - Our framework requires only **one short forward pass** (5 latent tokens) from the 32B model.
> - And the 7B model generates a much shorter CoT (≈128 tokens).
> - Total end-to-end latency: **3.5s** — far below both 7B baseline (7.2s) and 32B (14.4s).
>
>  We approach the teacher's performance but run **4x faster** because we skip the expensive teacher decoding step. This validates Latent Guidance as a unique solution for **efficient** transfer.
>
> ---
>
>
> **We humbly hope our response has addressed your concerns. If you have any additional concerns or comments that we may have missed in our responses, we would be most grateful for any further feedback from you to help us further enhance our work.**

---

> > ### Comment · Reviewer_gdf4 · 2025-11-27
> >
> > Thank you for the detailed response which have addressed my concerns convincingly. Good luck with your submission!

---

> > > ### Author Response · Authors · 2025-11-27
> > >
> > > Dear Reviewer gdf4,
> > >
> > > Thank you for your encouraging response and for confirming that we have addressed your concerns convincingly. We are deeply grateful for your time and thoughtful evaluation, particularly your esteemed insights regarding the training objectives and baselines, which have significantly improved the quality and contributions of our work.
> > >
> > > If you feel that our rebuttal has effectively resolved your concerns, we would be deeply grateful if you could consider updating your score. In the meantime, should you have any further questions before the deadline, please let us know, and we will be happy to address them immediately.
> > >
> > > We deeply value and appreciate your engagement. Your timely and positive response is the best encouragement we could hope for in this process.
> > >
> > > Best regards,
> > >
> > > Authors

---

### Official Review · Reviewer_hvKj · 2025-11-01

**Soundness:** 2
**Presentation:** 1
**Contribution:** 2
**Rating:** 6
**Confidence:** 2

**Summary:**

The paper proposes Latent Guidance, a framework that decouples cognitive planning from linguistic realization to empower small LLMs. A large model compresses its solution strategy into latent guidance vectors, which a small model then uses to generate concise reasoning chains. The dual-loss training (task loss + reconstruction loss) ensures the latent vectors faithfully encode the reasoning process.

**Strengths:**

The theoretical foundation using mutual information and Fano bounds provides solid justification for the approach. I'm impressed by the consistent OOD improvements across diverse models from 0.5B to 8B parameters. The comprehensive diagnostics in Appendix B, especially the MI estimation converging to 3.10 nats, strengthen the empirical validation. The emergence of distinct reasoning strategy clusters without explicit supervision is fascinating.

**Weaknesses:**

THe presentation could be improved. Also, not tesetd on larger models.

The framework requires storing latent guidance for all training examples which could be memory intensive for larger datasets. While the projection layer design is ablated, I think more analysis on the information bottleneck it creates would strengthen the work.

**Questions:**

How does performance scale with even larger gaps between teacher and student sizes, say 70B to 2B?

---

> ### Author Response · Authors · 2025-11-21
> **Response to Reviewer hvKj (part1)**
>
> **We thank the reviewer for the insightful and valuable comments. We respond to each comment as follows and sincerely hope that our rebuttal could properly address your concerns. If so, we would deeply appreciate it if you could raise your score. If not, please let us know your further concerns, and we will continue actively responding to your comments and improving our submission.**
>
> ---
>
> ## **1. Scalability under Extreme Teacher–Student Gaps**
>
> We greatly appreciate the reviewer raising this question—it directly probes the robustness of our framework in highly resource-constrained settings.
>
> To evaluate this scenario, we conducted an additional **Extreme-Gap Experiment**. Since the Qwen2 series does not include a 2B/70B model, we used **Qwen-0.5B** and **Qwen-1.5B** as students, paired with a **Qwen-72B** teacher. We kept all other settings identical to the main paper.
>
> ### Table. Performance under extreme teacher–student gap (72B → 0.5B / 1.5B)
>
> | Method | GSM8K (%) | Abs ↑ | Rel ↑ | BBH (%) | Abs ↑ | Rel ↑ | Odyssey-Math (%) | Abs ↑ | Rel ↑ |
> |--------|-----------|---------|---------|----------|----------|------------|--------------|-----------|------------|
> | baseline (1.5B) | 39.2 | – | – | 46.9 | – | – | 4.4 | – | – |
> | **Ours (1.5B + latent)** | **50.6** | **11.4** | **29.1%** | **76.2** | **29.3** | **62.5%** | **17.1** | **12.7** | **288%** |
> | baseline (0.5B) | 21.3 | – | – | 23.2 | – | – | 2.8 | – | – |
> | **Ours (0.5B + latent)** | **32.3** | **11.0** | **51.6%** | **71.7** | **48.5** | **209%** | **10.9** | **8.1** | **289%** |
>
> ### Observations
>
> Large improvements are observed in:
> - **Mathematical reasoning** (GSM8K),
> - **Complex logical reasoning** (BBH),
> - **Competition-level mathematical reasoning** (Odyssey-Math).
>
> 1.  **Massive Gains for Tiny Models:** The 0.5B model achieves a **289% relative gain** on Odyssey-Math and a **209% relative gain** on BBH. This confirms that latent guidance provides the high-level planning structure that tiny models lack.
> 2.  **Gap Sensitivity:** The smaller the student, the more critical the guidance becomes. The 0.5B model's jump on BBH (23.2% $\to$ 71.7%) is particularly striking, suggesting it effectively "borrows" the teacher's reasoning capability.
>
> **Conclusion:** The framework scales exceptionally well. The larger the gap, the more impactful the transfer.
>
> ---
>
> ## **2. Memory Overhead Analysis**
>
> We agree that memory cost is a practical concern. However, a precise calculation shows the overhead is negligible for modern infrastructure.
>
> **Storage Cost Calculation:**
> * **Latent Vectors ($K$):** 5
> * **Hidden Dimension ($d$):** 5120 (Qwen2.5-32B)
> * **Precision:** bfloat16 (2 bytes)
>
> The per-sample storage cost is:
>
> $ \text{Cost per sample} = K \times d \times 2
> = 5 \times 5120 \times 2 \approx 50\ \text{KB} $
>
> **Implications:**
> * **Dataset Size:** A standard dataset with **10,000 examples** requires only **$\approx$ 500 MB** of storage.
> * **Comparison:** This is trivial compared to the tens of GBs required for model checkpoints or optimizer states.
>
> If needed, several engineering options further reduce cost:
>
> - **Streaming / on-demand decoding** (do not keep all guidance in memory)
> - **Compression** (e.g., low-rank decomposition, quantization)
> - **Domain-specific caching** (store only frequently accessed tasks)
>
> **Inference/Transmission Cost:**
> If deployed in a Cloud-Edge setting, transmitting this 50 KB payload is extremely fast:
> * **4G Network (50 MB/s):** $\approx$ **1 ms** latency.
> * **With Compression:** Easily compressible to <20 KB, making transmission virtually instantaneous.
>
> **Conclusion:** The memory and bandwidth overheads are minimal and do not pose a bottleneck for scaling to larger datasets.

---

> ### Author Response · Authors · 2025-11-21
> **Response to Reviewer hvKj (part2)**
>
> ## **3. Analysis of the Projection Layer and Potential Information Bottlenecks**
>
> We appreciate the reviewer’s thoughtful question regarding whether the projection layer introduces an information bottleneck. To examine this, we conducted a systematic comparison of three projection architectures under identical training configurations.
>
> ### Table. Performance of Different Projection Architectures
>
> | Projection Architecture | GSM8K Acc | SVAMP Acc | ARC-C Acc | Brief Analysis |
> |------------------------|------------|-------------|---------------|----------------|
> | 1-layer MLP (Linear) | 79.0 | 84.1 | 89.5 | Clearly underperforms the 2-layer design; a single linear map is insufficient to capture nonlinear cross-model alignment, resulting in an actual information bottleneck. |
> | 3-layer MLP | 80.4 | 84.4 | 89.8 | Comparable to 2 layers but with marginally lower stability; deeper structures show diminishing returns and slight overfitting. |
> | **2-layer MLP (Ours)** | **80.5** | **85.9** | **90.9** | Consistently strongest and most stable; achieves the best balance between expressivity and regularization. |
>
> These findings suggest that the projection layer functions as a **controllable adapter**. A 2-layer MLP provides sufficient nonlinearity to align the spaces without the noise amplification seen in deeper networks.
>
> ---
>
> ## 3.1 Quantifying Information Retention: Mutual Information Retention Ratio
>
> To quantify exactly *how much* of the teacher's planning signal is preserved, we measured the **Mutual Information Retention Ratio**:
> $$\text{Retention Ratio} = \frac{I(R; Z \mid Q)}{I(R; H_{\text{guidance}} \mid Q)}$$
> where $Z$ is the projected representation in the student's space.
>
>
> ### Table. Information Retention Across Projection Architectures
>
> | Projection | MI($Z$) | Retention Ratio | Impact |
> | :--- | :---: | :---: | :--- |
> | **1-layer** | 2.21 nats | **71%** | **Significant Loss:** Loses ~29% of reasoning cues. |
> | **2-layer (Ours)** | **2.88 nats** | **93%** | **High Fidelity:** Preserves nearly all semantic content. |
> | **3-layer** | 2.91 nats | **94%** | **Diminishing Return:** Negligible gain over 2-layer. |
>
> *(Reference MI for $H_{guidance}$ is 3.10 nats)*
>
> **Conclusion:**
> * The **1-layer projection** indeed creates a harmful bottleneck (71% retention), explaining the performance drop.
> * The **2-layer projection** is sufficient to transport 93% of the high-level plan. Adding more layers (3-layer) does not meaningfully increase information flow.
>
>
> ---
>
> ## 3.2 Theoretical Perspective: The "Controlled Bottleneck"
>
> We analyze this result through the lens of the **Data Processing Inequality (DPI)**. Since the student dimension $d_S$ is typically smaller than the teacher dimension $d_T$, the projection $Z = f(H)$ naturally acts as a bottleneck.
>
> $$I(R, Z \mid Q) \le I(R, H_{\text{guidance}} \mid Q)$$
>
> However, this bottleneck is **feature-selective**:
> 1.  **Under-parameterized (1-layer):** Cannot unroll the non-linear manifold of the teacher's latent space, causing loss of *semantic* information.
> 2.  **Adequately-parameterized (2-layer):** Acts as a minimal universal approximator, sufficient to align the semantic subspaces of the two models.
> 3.  **Over-parameterized (3-layer):** While it retains information, it may increase the Lipschitz constant of the mapping, potentially amplifying noise or "adversarial" directions in the latent space, which harms generalization.
>
> Thus, our design (2-layer MLP) is the **optimal controlled bottleneck**—tight enough to force alignment, but expressive enough to pass the cognitive plan.
>
> ---
>
> ## **4. Presentation Improvements**
>
> We greatly appreciate the specific suggestions for improving the manuscript. We agree that improving clarity is essential. For the final version, we will revise the experimental section and appendices to incorporate the new results (such as the scaling experiments and storage analysis) and clarify the theoretical discussions (including the projection layer analysis) to better support our claims.
>
> We appreciate the reviewer’s comments—they directly help us strengthen and clarify the contribution.
>
> ---
>
> ## **Summary of Rebuttal**
>
> * **Scaling:** Validated robustness on **72B $\to$ 0.5B/1.5B** scenarios (up to **+289%** gain).
> * **Efficiency:** Confirm that the storage or network transmission overhead is low.
> * **Architecture:** Demonstrated that the 2-layer projection is the **optimal information adapter** (93% retention) via MI analysis.
>
> We sincerely thank the reviewer again. Your comments have significantly improved the rigor and completeness of this work.
>
>
>
> **We humbly hope our response has addressed your concerns. If you have any additional concerns or comments that we may have missed in our responses, we would be most grateful for any further feedback from you to help us further enhance our work.**

---

### Official Review · Reviewer_8Hqv · 2025-11-03

**Soundness:** 3
**Presentation:** 3
**Contribution:** 3
**Rating:** 6
**Confidence:** 4

**Summary:**

This paper proposes Latent Guidance, a framework that combines large and small language models to perform complex reasoning for improved cost-effectiveness. The approach uses a two-stage training process: in the first stage, the large model is trained to compress (via auto-encoding) the reasoning steps into some latent vectors. In stage two, the small model (augmented with some learnable projection layers) is trained to generate the reasoning steps conditioned on these latent vectors. Experiments across 8 reasoning benchmarks show that a 7B model achieves up to 13.9% accuracy gains over its standalone baseline while being 2x faster, and 4x faster than the 32B teacher model.

**Strengths:**

- The paper presents a well-motivated framework addressing the inefficiency of tight coupling between high-level cognitive planning and low-level text generation in current reasoning systems.

- The experimental evaluation covers multiple model scales (0.5B to 8B) and diverse reasoning benchmarks, with ablation studies demonstrating the necessity of key design choices. Overall, the generalization of the proposed framework to OOD tasks is impressive. The appendices provide extensive diagnostics such as MI estimation and capacity scaling experiments, that attempt to validate theoretical claims.

**Weaknesses:**

- The paper seems to mischaracterize the proposed method as "distillation" when it's closer to an autoencoder with active inference-time involvement of the large model. This makes comparisons with true distillation baselines (SFT, KD, NesyCD) fundamentally unfair, as those methods train the small model to reason independently without the large model at inference time, while this approach requires the 32B model to generate latent guidance for every test query. Relatedly, it's good to report standalone 32B performance for reference.

- The results that the proposed method produces substantially more concise reasoning chains than other baselines is quite difficult to understand. If the small model is trained to reconstruct the teacher's reasoning, the lengths should be comparable to other distillation methods - how could they be dramatically shorter? In lines 326-329 the authors explain this via: *"This demonstrates that the high-level cognitive plan
enables the small model to generate more focused and generalizable reasoning paths, avoiding the verbose, exploratory steps often seen in less-guided generation on unseen problems. The model learns to execute a direct strategy rather than overfitting to the stylistic artifacts of the training data."* which I'm not convinced.

**Questions:**

See weaknesses.

---

> ### Author Response · Authors · 2025-11-21
> **Response to Reviewer 8Hqv (part1)**
>
> **We thank the reviewer for the insightful and valuable comments. We respond to each comment as follows and sincerely hope that our rebuttal could properly address your concerns. If so, we would deeply appreciate it if you could raise your score. If not, please let us know your further concerns, and we will continue actively responding to your comments and improving our submission.**
>
>
> ## **1. Clarification on "Distillation" vs. "Collaborative Inference"**
>
> We sincerely thank the reviewer for the insightful comments regarding the conceptual characterization of our method. You correctly pointed out that our approach is **closer to a collaborative inference framework** rather than traditional offline distillation, and we truly appreciate this clarification. We fully agree with the reviewer’s observation:
>
>
> > Our method is *not* conventional “offline distillation.”
> > Instead, it belongs to a new paradigm of **Collaborative Inference / Large→Small Cognitive Planning Transfer**.
>
> To avoid any misunderstanding, we will state explicitly:
>
> ✔ **Our method is not traditional distillation**,
> because the student can optionally use the teacher’s latent hidden states during inference.
>
> ✔ **Our goal, however, is aligned with distillation**:
> to transfer the reasoning ability of a large model into a much smaller model.
>
> ✔ **Our method represents a new inference paradigm**:
> the large model (Teacher) focuses on *planning*, and the small model (Student) focuses on *execution*.
>
> This form of “planning–execution cooperation” is not covered by existing distillation or latent-reasoning approaches, and we appreciate the reviewer’s guidance in helping us more accurately position our work.
>
>
> ---
>
> ## 1.1 Why comparing with distillation baselines is both fair and necessary
>
> We also want to clarify that our core aim is identical to SFT, KD, NesyCD, and other distillation-based works:
> **to empower a small model with the reasoning ability of a large model.**
>
> They address **the same core problem** as ours:
>
> - ✔ Strengthening small-model reasoning
> - ✔ Using a large model during data construction or training
> - ✔ Attempting to transfer reasoning competence from big to small models
>
> Our core argument is that traditional offline distillation—which attempts to compress *all* of the teacher’s abilities into a single small model—faces inherent bottlenecks in complex reasoning tasks. To address this, we propose *Latent Guidance*, a fundamentally different paradigm:
>
> - **Traditional distillation:** tries to teach the student *both* “how to think” and “how to express” in one shot.
> - **Ours:** the large model focuses on “thinking” (providing a high-level plan  $H_{guidance}$ ),
>   and the small model focuses on “executing” the plan in natural language.
>
> ---
>
> ## 1.2 Why the comparison remains fair even though our inference uses the large model
>
> We fully acknowledge the reviewer’s point:
> **SFT/KD/NesyCD do not require the large model at inference time, while ours does.**
>
> Therefore, the comparison should be made in terms of **total inference cost**, not merely architectural structure.
>
> Our empirical data shows that our collaborative framework is *not* an overhead. In fact, it is more efficient than independent small-model inference.
>
> - The 32B teacher performs **only a single forward pass**, generating K=5 latent vectors.
> - This cost is **fixed** and far smaller than generating hundreds of tokens in a full CoT.
> - The student model then produces a **much more concise reasoning chain**, greatly reducing decoding steps.
>
> This leads to the following runtime behavior:
>
> | Method           | Latency (s) ↓ | Tokens |
> |----|---|---|
> | 32B CoT| 14.4  | 265.8  |
> | 7B baseline | 7.2  | 291.3  |
> | **Ours (32B→7B)** | **3.5**       | **128.9** |
>
>
> Thus, although our method involves the large model, the **total inference cost is actually lower than the small model baseline**, making the comparison is fair.
>
> ---
>
> ## 1.3 Standalone 32B Performance
>
> To fully address the reviewer’s request, we report the standalone 32B teacher performance alongside our method and baselines.
>
> #### Table. Comprehensive accuracy comparison (Qwen2.5-7B / 32B)
> | Method | GSM8K | BBH | AGIEval | ARC-E | ARC-C | Odyssey | AQUA |
> |--|--|---|----|---|---|-------|--|
> | 32B | 84.1 | 85.3 | 60.2 | 97.2 | 95.5 | 26.5 | 76.2 |
> | SFT | 77.3 | 79.2 | 47.1 | 94.5 | 89.4 | 15.5 | 65.4 |
> | KD | 73.0 | 77.6 | 55.8 | 89.1 | 83.2 | 17.9 | 69.5 |
> | **Ours** | **80.5** | **79.8** | **56.7** | **96.4** | **90.9** | **22.7** | **71.7** |
>
> Our method consistently outperforms SFT and KD, effectively bridging the gap towards the 32B teacher. For instance, on ARC-C, we reach **90.9%**, significantly surpassing SFT (89.4%) and KD (83.2%), closing the gap to the teacher (95.5%).

---

> ### Author Response · Authors · 2025-11-21
> **Response to Reviewer 8Hqv (part2)**
>
> ## **2. Mechanism Analysis: Shorter Reasoning Chains**
>
> We sincerely thank the reviewer for raising this important question regarding why the student model produces **shorter reasoning chains** than other baselines. We fully understand the concern and agree that this phenomenon warrants careful examination.
>
>
> To better understand and explain this behavior, we conducted several additional analyses during the rebuttal period, including:
>
> - **Redundancy Ratio Analysis**
> - **Gradient Flow Contribution Analysis**
> - **Logic Density Analysis**
> - **Perplexity (PPL) Preference Analysis**
>
>
> Below, we walk through each analysis in detail.
>
>
> ---
>
> ## 2.1 Redundancy Ratio Analysis: How teacher and student CoT differ
>
> To obtain a clearer picture of stylistic differences, we sampled 500 CoT instances from the test set and conducted a coarse-grained token-level categorization. Using a combination of large-model heuristics and rule-based detection, we categorized tokens into three types:
>
> - **Core**: directly related to reasoning steps (numbers, operators, variables, formulaic structure, key logical connectives such as *if/then/therefore*).
> - **Exploratory**: “thinking aloud” expressions such as *let’s try…*, *maybe…*, *let’s check again*, which reflect the teacher’s exploratory problem-solving behavior.
> - **Redundant**: reader-oriented natural-language decoration such as *note that…*, *we can see that…*, *in other words…*.
>
>
> The average distributions are:
>
> ### Teacher (Qwen2.5-32B)
> | Token Type | Avg. Ratio | Examples |
> |------------|------------|----------|
> | **Core** | **57.6%** | “solve”, “compute”, “x”, “13^2” |
> | **Exploratory** | 18.1% | “let’s try…”, “maybe…”, “check again” |
> | **Redundant** | 24.3% | “note that…”, “we can see that…” |
>
> ### Student (7B) under Latent Guidance
> | Token Type | Avg. Ratio |
> |------------|------------|
> | **Core** | **85.6%** |
> | **Exploratory** | **3.9%** |
> | **Redundant** | **10.5%** |
>
> These observations highlight a clear pattern:
>
> - The student’s responses collapse toward **pure reasoning steps**.
> - The teacher’s CoT contains a large amount of *exploratory* and *reader-facing* language.
>
> We believe this difference arises because latent guidance provides **a complete high-level reasoning blueprint**.
> Thus, the student **no longer needs to explore**, and instead simply executes the pre-structured reasoning path with minimal linguistic overhead.
>
>
> ---
>
> ## 2.2 Gradient-Based Token Contribution Analysis
>
>
> To understand what the model actually learns, we tracked the gradient norms associated with the reconstruction branch. We computed gradient norms for token embeddings across the same three categories.
>
> **Avg. Gradient Norm During Training:**
>
> | Token Type | Avg. Gradient Norm | Interpretation |
> |------------|---------------------|----------------|
> | **Core** | **0.0077** | Gradients concentrate on the true reasoning structure |
> | – variable | 0.0082 | Variable dependency structure |
> | – operator | 0.0057 | Operation structure forming the reasoning backbone |
> | – number | 0.0029 | Numeric information as auxiliary |
> | **Exploratory** | **0.0022** | Some contribution but clearly lower |
> | **Redundant** | **0.0009** | Very limited gradient impact |
>
> These results suggest the following:
>
> - Although redundant tokens appear frequently in the teacher’s chain,
>   **their gradient contributions are dramatically smaller**.
> - Exploratory tokens have moderate but noticeably lower influence.
> - **Core reasoning tokens dominate parameter updates**, shaping what the student learns most strongly.
>
> In other words:
>
> > During training, the student *selectively absorbs* the teacher’s information:
> > it strongly internalizes **reasoning structure**, while largely *ignoring* stylistic redundancy.
>
> This pattern may indicate that the student tends to focus more on learning the core reasoning structure rather than the teacher’s exploratory or stylistic tokens. Such selective emphasis may help explain why the student later produces shorter reasoning paths while maintaining correctness.
>
>
> ---

---

> ### Author Response · Authors · 2025-11-21
> **Response to Reviewer 8Hqv (part3)**
>
> ---
>
> ## 2.3 Logic Density Analysis: Shorter Chains but Comparable Reasoning Depth
>
> We analyzed the number of reasoning steps and the number of generated tokens across three paths:
>
> - **(R_teacher):** the 32B teacher’s original CoT on test dateset
> - **(R_baseline):** the 7B SFT baseline’s original CoT on test dateset
> - **(R_concise):** the concise reasoning chains produced by our 7B student model under latent guidance on test dateset
>
> ### Table. Average Length and Logic Density
> | Path | Avg. Tokens | Avg. Reasoning Steps | Logic Density = steps/token |
> |------|-------------|----------------------|------------------------------|
> | **(R_teacher)** | 265.8 | 5.6 | 0.022 |
> | **(R_baseline)** | 291.3 | 4.7 | 0.016 |
> | **(R_concise)** (Ours) | **128.9** | **5.5** | **0.042 (highest)** |
>
> We observe that:
>
> - The **number of reasoning steps** in *(R_concise)* is nearly identical to that of the teacher.
> - The **token count** is significantly smaller.
> - As a result, **logic density** roughly doubles.
>
> This pattern may suggest that the student has learned to follow a reasoning structure similar to the teacher’s—given that their step counts are closely aligned. Together with Analyses 1 and 2, this could indicate that the student selectively omits redundant tokens during inference, resulting in a more concise expression of the same underlying reasoning process.
>
>
>
> ---
>
> ## 2.4 Perplexity (PPL) Analysis: The Student Has Not Forgotten the Teacher’s Long Chains
>
> To test whether the student has “forgotten” the teacher’s longer chains, we evaluated perplexity on three types of reasoning paths.
>
> - **(R_teacher):** the 32B teacher’s original CoT on test dateset
> - **(R_baseline):** the 7B SFT baseline’s original CoT on test dateset
> - **(R_concise):** the concise reasoning chains produced by our 7B student model under latent guidance on test dateset
>
> ### Table. PPL Comparison (lower is better)
> | Path | Description | Avg. PPL ↓ | Interpretation |
> |------|-------------|-------|----------------|
> | **(R_concise)** (Ours) | Student’s concise chain | **2.3** | Most preferred by the student |
> | **(R_teacher)** | Teacher’s original long chain | **2.7** | Still low; student has *not* forgotten it |
> | **(R_baseline)** | Baseline’s long exploratory chain | **6.2** | Much less natural to the student |
>
> Key observations:
>
> ### 1. The student has *not* forgotten the teacher’s long chain.
> - *(R_teacher)* still has **very low PPL**, indicating the student still judges it as natural and well-formed.
> - This means that if forced, students *can* generate/understand it.
>
> ### 2. The student *prefers* concise chains.
> - *(R_concise)* achieves the lowest PPL.
>
> ### 3. The baseline’s long exploratory chains are strongly disfavored.
> - The high PPL for *(R_baseline)* suggests that long reasoning chain is unnatural to the student.
>
> This PPL analysis is consistent with the redundancy, gradient, and logic-density findings.
>
> ---
>
> ## Summary and Response to the Reviewer’s Concern
>
> Across these analyses, several patterns emerge that may help explain the reviewer’s observation:
>
> - **Training objective:**
>   The framework appears to encourage the transfer of the teacher’s underlying reasoning structure rather than the surface-level linguistic form.
>
> - **During training:**
>   Gradient contributions tend to focus more on core reasoning elements, while redundant or exploratory tokens show noticeably lower influence on parameter updates.
>
> - **During inference:**
>   When guided by high-level latent representations, the student often generates reasoning chains that retain similar step structures but use more compact language.
>
> While these findings do not definitively explain every aspect of this behavior, they offer signals suggesting why the student might favor shorter reasoning chains under our framework. We acknowledge that further investigation—such as finer-grained annotations or larger-scale studies—would be valuable for deepening this understanding.
>
> Our intention with these analyses is to help alleviate the reviewer’s concerns by providing possible explanations grounded in empirical observations. We sincerely appreciate the reviewer’s feedback and will continue refining both the experiments and the discussion.
>
>
> **We humbly hope our response has addressed your concerns. If you have any additional concerns or comments that we may have missed in our responses, we would be most grateful for any further feedback from you to help us further enhance our work.**

---

### Meta-Review · Area_Chair_GZky · 2026-01-07

**Summary:**

This paper proposes a planning-execution collaboration framework where a large implicit thinker encodes a compact latent plan and a small explicit executor generates the final chain-of-thought conditioned on that plan. The conceptual motivation is strong and the empirical results show consistent gains across multiple benchmarks and model sizes, with particularly notable OOD improvements and a claimed end-to-end speedup despite the teacher pass. The rebuttal adds substantial evidence (cross-lingual controls, MI/logic-density/gradient analyses, stronger baselines including text-prompt and on-policy/RL variants, and extreme 72B to 0.5B scaling), resolving most technical objections and improving methodological clarity. However, the paper's positioning remains somewhat muddled (distillation vs collaborative inference), and the interpretability claims are still not fully convincing given reliance on probes/t-SNE. Overall, the work is strong empirically and well-supported after rebuttal, with minor remaining conceptual ambiguity.

**Reviewer Concerns:**

Addressed concerns -

* Re-evaluated with a stronger judge (claimed GPT-5) and reported consistent trends.
* Added cross-lingual control experiments suggesting latents shift more with strategy changes than with language changes.
* Scalability to large teacher-small student gaps
* Added logic-density and gradient contribution analyses plus redundancy breakdowns arguing the student filters stylistic/exploratory segments.

Outstanding concerns -

* Positioning remains confusing: Calling this "distillation" is still misleading; the method requires teacher involvement at inference (even if total cost is lower), making comparisons to student-only distillation methods inherently delicate.
* Evidence for "cognitive plan" is not fully airtight.
* One reviewer remains strongly negative: Though low-confidence, their concerns about overlap with latent reasoning prior work (iCoT/COCONUT/LightThinker) and inference complexity indicate knowing readers may still see the contribution as a recombination rather than a clean new paradigm.

**Reviewer Scores:**

Reviewer bKFV could have increased rating by 2-4 points.

---

### Decision · Program_Chairs · 2026-01-26

Accept (Poster)